# A Performance-Based Approach to Quantify Atmospheric River Flood Risk

Corinne Bowers[1], Katherine A. Serafin[2], and Jack Baker[1]

[1]Department of Civil and Environmental Engineering, Stanford University, Stanford, CA, USA
[2]Department of Geography, University of Florida, Gainesville, FL, USA

**Correspondence:** Corinne Bowers (cbowers@stanford.edu)

**Abstract.**

Atmospheric rivers (ARs) are a class of meteorologic phenomena that cause significant precipitation and flooding on the US West Coast. This work presents a new Performance-based Atmospheric River Risk Analysis (PARRA) framework that adapts existing concepts from probabilistic risk analysis and performance-based engineering for application in the context of AR-driven fluvial flooding. The PARRA framework is a chain of physically based models that link the atmospheric forcings, hydrologic impacts, and economic consequences of AR-driven fluvial flood risk together at consistent "pinch points." Organizing around these pinch points makes the framework modular, meaning models between pinch points can be updated without affecting the rest of the model chain, and it produces a probabilistic result that quantifies the uncertainty in the underlying system states. The PARRA framework can produce results beyond analyses of individual scenario events and can look towards prospective assessment of events or system changes that have not been seen in the historic record.

The utility of the PARRA framework is demonstrated through a series of analyses in Sonoma County, California. Individual component models are fit and validated against a historic catalog of AR events occurring from 1987-2019. Comparing simulated results from these component model implementations against observed historic ARs highlights what we can learn about the drivers of extremeness in different flood events by taking a probabilistic perspective. The component models are then run in sequence to generate a first-of-its-kind AR flood loss exceedance curve for Sonoma County. The prospective capabilities of the PARRA framework are presented through the evaluation of a hypothetical mitigation action. Elevating 200 homes, selected based on their proximity to the Russian River, was sufficient to reduce the average annual loss by half. While expected benefits were minimal for the smallest events in the stochastic record, the larger, more damaging ARs were expected to see loss reductions of approximately $50-$75 million per event. These results indicate the potential of the PARRA framework to examine other changes to flood hazard, exposure, and vulnerability at the community level.

## 1  Introduction

Atmospheric rivers (ARs) are long ($>2000\,\mathrm{km}$) and narrow (500-1000 km) corridors of strong horizontal water vapor transport, with water concentrated mostly in the lowest 3 km of the atmosphere (Ralph et al., 2018). ARs are the primary vector for moving moisture from the tropics to the midlatitudes, responsible for up to 90% of longitudinal water transport while covering

only 10% of the earth's surface (Zhu and Newell, 1998). ARs are crucial to the stability of California's water resources: in just a hundred hours of rain per year they can deposit up to half of the state's annual water supply (Lamjiri et al., 2018). However, this gift comes at a price. ARs cause well over three-quarters of all extreme precipitation events in California and over 90% of the state's record floods (Lamjiri et al., 2018), leading to almost $660 million in average annual losses (Corringham et al., 2019).

One particularly devastating AR event was the Great Flood of 1862. Central California received over ten feet of precipitation in just 43 days between December 1861 and January 1862, and cities from San Francisco to San Diego set precipitation records that still stand today. Based on this catastrophe, the US Geological Survey (USGS) created a hypothetical AR scenario named ARkStorm with a return period of approximately 500–1000 years (Porter et al., 2011). The study found that California's flood control infrastructure, mostly built to withstand 100- or 200-year return period events, was woefully underprepared for this disaster scenario, and that much needed to be done in terms of both mitigation investment and emergency planning to prevent an economic catastrophe. The ARkStorm report concluded with a call for an "end-to-end stochastic model of severe weather, physical impacts, and socioeconomic consequences" (Porter et al., 2011).

This paper presents a novel performance-based risk analysis framework that integrates meteorological, hydrological, and engineering models to assess AR-induced fluvial flood risk. The Performance-based AR Risk Analysis (PARRA) framework probabilistically models ARs from inception all the way to economic consequences, as first outlined a decade ago in the ARkStorm report. The core goal of the PARRA framework is to provide a conceptual outline to link research related to the atmospheric forcings, hydrologic impacts, and economic consequences of AR-induced flood risk. It is (a) physically based, meaning it supports precise and consistent modeling of the sequence of processes from inception to impacts; (b) modular, meaning individual component models can be modified without affecting the rest of the sequence; (c) probabilistic, meaning the full uncertainty quantification at each step is carried through the model sequence to assess confidence in the final results; and (d) prospective, meaning the framework can be used to assess "what-if" questions about events that have not yet occurred. These characteristics set it apart from previous scenario-based and statistical analyses of AR-driven losses and allow for a more comprehensive evaluation of AR-induced fluvial flood risk.

## 1.1 Disciplinary Context

Prediction of flood damage and loss due to ARs requires disciplinary expertise spanning meteorology, hydrology, engineering risk analysis, and more. Most previous research modeling ARs has focused on two pathways: the first linking atmospheric forcings to hydrologic impacts, and the second linking hydrologic impacts to economic consequences.

The first physical process to consider is the transformation of atmospheric phenomena into precipitation and runoff. Considerable effort has been invested into understanding the climatology of ARs, through collection of meteorological field data (Lavers et al., 2020), improvements to existing numerical weather prediction models (Nardi et al., 2018; Martin et al., 2018), and intercomparison between AR detection algorithms (Shields et al., 2018). Researchers have described the particular climatology of ARs affecting California and the US West Coast (Waliser and Guan, 2017; Guirguis et al., 2018). Characterizing these features and their spatial and temporal distributions allows us to better connect AR events with their hydrological impacts,

namely extreme precipitation (Chen et al., 2018; Huang et al., 2020) and runoff (Konrad and Dettinger, 2017; Albano et al., 2020).

Translating from hydrologic impacts to economic consequences is generally a multi-step process, and in the literature there are both specific models for portions of the process and multi-model sequences. Hydrologic routing software such as HEC-HMS (USACE, 2020) determine the shape of the event hydrograph at a given point along the river based on a precipitation event, and hydrodynamic solvers such as HEC-RAS (Brunner, 2020) or LISFLOOD-FP (Bates and De Roo, 2000) generate 2D maps of the resulting inundation. These tools are generally designed for practitioners to assess the consequences of the "100-year event," which is the hydrological event with a 1% annual probability of occurrence. The 100-year event is a term with a long history in planning and engineering design and is generally set as the standard that flood control infrastructure must be built to withstand. In the US, the Federal Emergency Management Agency (FEMA) has been defining flood risk in these terms for decades, through the Federal Insurance Rate Maps (FIRMs) that delineate the 100-year floodplain and through their open-source loss estimation software Hazus-MH (FEMA, 2020). Other regional flood loss assessment tools include HEC-FIA (USACE, 2018) and HEC-FDA (USACE, 2014), both from the US Army Corps of Engineers, and FloodFactor (Bates et al., 2020), a commercial product from First Street Foundation.

## 1.2 Methodological Frameworks

Some studies have gone beyond domain-specific solutions to capture the entire chain of processes from atmospheric events to economic loss (Dominguez et al., 2018; Felder et al., 2018; Porter et al., 2011). However, these previous works have focused on the consequences of one specific disaster scenario rather than the full spectrum of AR flood risk in a given location. Corringham et al. (2019) subsequently connected ARs directly to their economic impacts and produced the first estimate of total flood loss attributable to ARs in the Western US. While this was an important step in understanding the consequences of these storms, it included no physical modeling of the processes connecting ARs to losses. The framework proposed here provides a structure to overcome these challenges. The PARRA framework is an organized sequence of component models that connect AR occurrence to the damage and losses that result from AR-driven flooding. Capturing these physical connections with disciplinary models of atmospheric forcings, hydrologic impacts, and economic consequences allows for a greater depth of understanding of the underlying phenomenological drivers.

The idea of "model chains" or "model sequences" is born from probabilistic risk assessment, a field that originated in the regulation of nuclear reactors (NRC, 1975). Probabilistic risk assessment frames risk as a function of likelihood times consequence. The overall risk at a site is the product of the likelihood and consequence of a given outcome, integrated over all possible outcomes in the event space (Baker et al., 2021). The report that first introduced probabilistic risk assessment relied on logic trees to organize these possible outcomes (NRC, 1975). Since then, more complex models and combinations of models have been adopted to improve the representation of the event space. The use of multi-model sequences is now common in natural disaster risk assessment and has grown in popularity as a way to address flood risk (Apel et al., 2004; Felder et al., 2018; Uhe et al., 2020). The insurance industry in particular has embraced model chains to create catastrophe risk models for an array of natural hazards (Pinelli and Barbato, 2019). While the insurance models prove the viability and utility of the

proposed concept, they are often limited by legal or proprietary constraints and so are not useful for generating public data or performing research. Therefore the PARRA framework is defined using the language of performance-based engineering, which focuses on the variables organizing the model sequence rather than the component models themselves.

Performance-based engineering (PBE) refers to designing a system to meet a target performance objective with a specified reliability rather than satisfying prescriptive requirements. Rather than defining the models that must be linked together, PBE frameworks are designed around "pinch points" where only a small amount of information must be passed from one step to the next (Garrick, 1984). The first PBE framework was introduced to estimate losses to buildings in future earthquakes (Krawinkler, 1999). The success in earthquake engineering led to development of PBE frameworks for other hazards, including wind (Ciampoli et al., 2011), hurricanes (Barbato et al., 2013), fire (Guo et al., 2013), and corrosion (Flint et al., 2014). Condensing the models down to a reduced set of pinch points at designated steps within the model chain dramatically improves the flexibility of the framework and helps organize research efforts into interchangeable and modular components.

## 2 Framework Description

The PARRA framework is composed of six *component models* connected by *pinch points*. Component models are the physical processes that make up the chain of events from AR occurrence to flood loss; pinch points are the points in the modeling chain where information is passed between component models. We first develop the overall structure of the framework and then introduce the specific pinch points and component models.

At a fundamental level the PARRA framework is an implementation of the law of total probability, which states that $P(A) = \sum_{i=1}^{n} P(A|B_i)\,P(B_i)$. In this equation $P(A|B_i)$ represents the conditional probability of event $A$ given that event $B_i$ has occurred and $P(B_i)$ represents the probability of event $B_i$ out of some set of $n$ mutually exclusive, collectively exhaustive events. Summing these probabilities over all possible instances of $B_i$ gives us the total probability of event $A$.

Equation 1 modifies the statement of the law of total probability to better fit the context of natural hazard assessment.

$$\lambda(DV > x) = \int_{AR} P(DV > x\,|\,AR) \cdot \lambda(AR)\,dAR \tag{1}$$

where $\lambda(DV > x)$ is the rate of the decision variable $DV$ exceeding some specified threshold $x$, i.e., how frequently losses exceed \$$x$ dollars; $P(DV > x\,|\,AR)$ is the probability of $DV$ exceeding $x$ conditioned on an inducing AR event; and $\lambda(AR)$ is the occurrence rate of that inducing event. The right side of the expression is integrated over all possible inducing events in the sample space. We evaluate $\lambda(DV > x)$ at a range of $x$ values to obtain the loss exceedance curve, which is developed further in Sect. 4.3.

We first replace the generic variables with new variables representing pinch points, which we elaborate on later in this section. $B$ becomes the atmospheric river event $AR$ and $A$ becomes the decision variable $DV$. $P(DV > x)$ is the complement of the cumulative distribution function for $DV$, starting at 100% probability of exceedance for low values of $x$ and moving to a probability of zero as $x$ increases. $P(DV > x\,|\,AR)$ represents the probability of the decision variable $DV$ exceeding some threshold value $x$ conditioned on the inducing event $AR$.

125    We then transform the summation into an integral and move to calculating the occurrence rate $\lambda$, which represents a continuous state variable rather than the probability $P$ of a discrete event. Probabilities are defined with respect to predetermined time periods, and the probability of seeing an AR event in the next week, month, or year are all different quantities. Calculating the occurrence rate $\lambda$ offers similar information about the underlying phenomenon of interest (AR event frequency) without imposing an arbitrary time limitation.

Equation 1 forms the theoretical basis for the construction of a performance-based probabilistic framework for ARs. However, it is difficult to explicitly calculate $P(DV > x \,|\, AR)$ in order to evaluate the equation, because the pathway from ARs to flood loss involves many complex atmospheric, hydrologic, and economic processes. Therefore we expand this probability statement by defining intermediary *pinch points* that decompose the calculations into a series of *component models*.

Pinch points are the "links" in the model chain that mark the end of one physical process and the start of another, depicted
by the arrows in Fig. 1. The pinch points in the PARRA framework are: $AR$, atmospheric river (some measure of intensity for a specific AR event); $PRCP$, precipitation (accumulated rainfall at the location or watershed of interest); $HC$, antecedent hydrologic conditions (some measure of the preexisting water balance within the watershed); $Q$, streamflow (the inflow hydrograph for the study area of interest); $INUN$, inundation (the height of water at buildings or locations of interest within the study area); $DM$, damage measure (damage ratios at buildings or locations of interest within the study area); and $DV$,
decision variable (some metric of impact or consequence for the study area). Each of these pinch points is explained in greater detail in Sect. 2.1.

Component models are representations of discrete physical processes in the series of events connecting ARs to flood losses, depicted by white boxes in Fig. 1. The six component models are: AR occurrence/magnitude, precipitation, hydrologic routing, inundation, depth-damage curves, and loss estimation.

We combine the pinch points and component models to form Eq. 2, which represents the PARRA framework in its entirety.

$$
\begin{aligned}
\lambda(DV > x) = \int \int \int \int \int \int \; & P(DV > x \,|\, DM) \cdot \\
& f(DM|INUN) \cdot \\
& f(INUN|Q) \cdot \\
& f(Q|PRCP, HC) \cdot f(HC) \cdot \\
& f(PRCP|AR) \cdot \lambda(AR) \\
& dDM \; dINUN \; dQ \; dHC \; dPRCP \; dAR
\end{aligned}
\tag{2}
$$

where variables $AR$, $PRCP$, $HC$, $Q$, $INUN$, $DM$, and $DV$ represent pinch points and the conditional probability expressions represent component models. The component models of the form $f(Y|X)$ are conditional probability density functions that describe the distribution of results from numerical analyses. The component model $P(DV > x \,|\, DM)$ measures the probability of pinch point $DV$ exceeding the loss threshold $x$ conditioned on $DM$. The PARRA framework is executed by starting

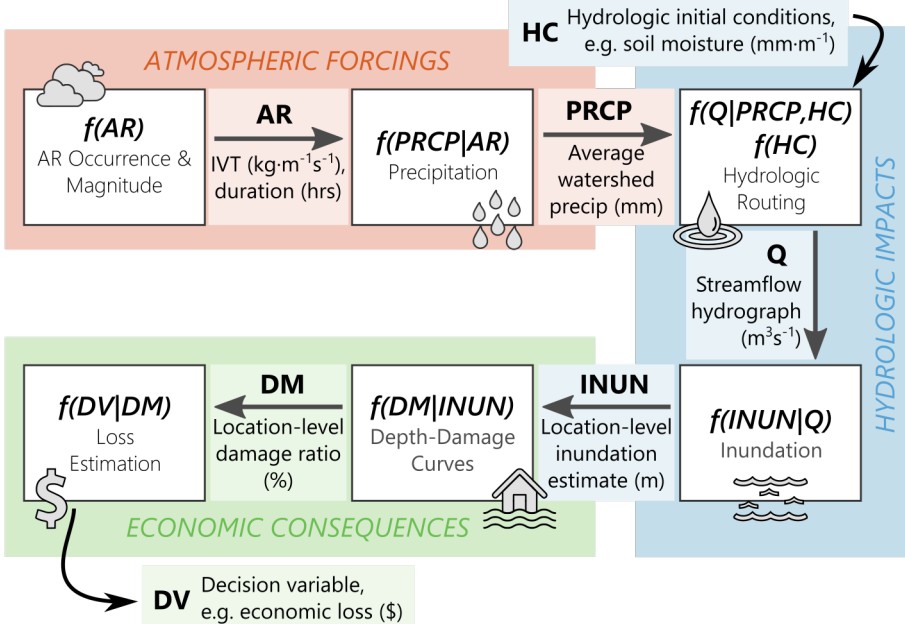

**Figure 1. PARRA framework flowchart.** Graphical depiction of the PARRA framework, as presented mathematically in Eq. 2. White boxes represent component models. Arrows represent pinch points: an arrow pointing towards a box indicates a required component model input, and an arrow coming out of a box indicates a component model output. The background colors broadly represent existing research domains.

with the outermost integration in the equation and moving inward, as each component model is conditioned on the one(s) preceding it in the model chain. This equation is also represented visually in Fig. 1.

In practice, Eq. 2 and other similar performance-based frameworks often cannot be reduced to a closed form equation. Therefore they are typically solved through Monte Carlo simulation. Each one of the component models is implemented as a numerical analysis that produces a best-fit projection for the next pinch point variable in the model sequence plus some characterization of the uncertainty in that projection. Rather than explicitly parameterizing the uncertainty at every step, we define empirical relationships based on the overall distribution of the historic record, then generate Monte Carlo samples to produce multiple stochastic realizations of each pinch point variable. These stochastic realizations of potential system states are propagated through the model chain to build an empirical distribution of expected loss. A more in-depth discussion of framework implementation and Monte Carlo simulation can be found in Sect. 5.1.

## 2.1 Pinch Point Variables

The pinch points presented in Sect. 2 are conceptual descriptions of the intermediate system states between AR occurrence and flood loss where only a limited amount of information must be transferred to the next step. Pinch point variables are low-dimensional numerical vectors representing the information recorded at each pinch point (Garrick, 1984). The following

paragraphs expand upon the conceptual pinch points and introduce the specific dimensions and measurement units that are used in this paper for each pinch point variable.

The pinch point variable representing an atmospheric river event ($AR$) is characterized as a vector with two elements: the maximum recorded integrated water vapor transport (IVT) ($\text{kg} \cdot \text{m}^{-1}\text{s}^{-1}$) and the duration (h) of sustained IVT exceeding 250 $\text{kg} \cdot \text{m}^{-1}\text{s}^{-1}$. These were chosen as metrics of interest because of their connection to impacts. Based on maximum IVT and duration, the bivariate AR intensity scale proposed by Ralph et al. (2019) ranks ARs from 1–5 to qualitatively summarize their expected severity (from weak to exceptional) and potential consequences (from beneficial to hazardous). Category 1 ARs are classified as primarily beneficial storms, replenishing the water supply without causing adverse effects. Category 5 ARs are classified as primarily hazardous with a high likelihood of flooding and damage.

Precipitation ($PRCP$) measures the amount of water released by the AR as measured at the ground surface, generally recorded as a height or depth. Only precipitation associated with ARs is included in this analysis. Precipitation is variable in both space and time and can potentially be summarized in several ways. In this work we define the precipitation pinch point variable as a scalar value representing the storm-total accumulated rainfall averaged across the spatial extent of the upstream watershed. This simplification could be modified or eliminated in future implementations of the precipitation component model, as addressed further in Sect. 5.3.

Hydrologic conditions ($HC$) refer to the saturation state before the storm, i.e., how much water is already present in the watershed system. Previous research has found antecedent soil moisture to be an important factor in predicting which precipitation events will lead to flood events (Cao et al., 2019). Therefore the pinch point variable representing antecedent hydrologic conditions is a scalar value measuring the average soil moisture in the upstream watershed.

The streamflow hydrograph ($Q$) is the timeseries of flow measurements vs. time recorded at the study area inlet for the duration of the AR event. Instead of storing this entire vector of flow vs. time values, we parameterize the hydrograph with three variables: $Q_p$, peak streamflow ($\text{m}^3\text{s}^{-1}$); $t_p$, time to peak streamflow (h); and $m$, a unitless shape parameter. This parameterization process is discussed in more detail in Sect. 3.4.

Inundation ($INUN$) is the surface water depth at locations of interest within the study area. These values should be zero or positive-valued only, as they represent a height of water above the ground surface. The vector of heights at $N$ locations of interest is stored for the next component model.

The damage measure ($DM$) is defined in this work as a damage ratio, or the expected cost to repair a building divided by the total value of that building. The damage ratio is assumed to be purely a function of water depth with respect to the first floor elevation. The result of the depth-damage calculation is a length $N$ vector of the same size as $INUN$ where 0 signifies no damage and 1 signifies a cost of repair equal to the value of the building.

Finally, the decision variable ($DV$) is some actionable measure of AR impacts. In this work we define $DV$ as household-level monetary losses; however, $DV$ could alternatively represent any other metric that is calculated as a function of the damage measure $DM$, such as the number of displaced persons or the time to full recovery. The expected loss for each structure is the value of exposed assets, namely building and contents valuations, multiplied by the damage ratio. The result is a length $N$ vector that represents the expected loss for each location of interest.

## 2.2 Component Models

The pinch points in the PARRA framework are linked by component models, or representations of discrete physical processes. Each component model generates an expected distribution of values for the next pinch point variable in the sequence conditioned on the value(s) preceding it. It is important to note that, excepting the hydrologic routing model $f(Q|PRCP, HC)$, all models are conditioned on only one variable. The hydrologic routing model differs from the others because, like the event characteristics ($AR$), the antecedent hydrologic conditions ($HC$) are framework inputs provided by the user to represent an initial system state. All other pinch point variables represent calculated variables. Conditioning on a minimal number of variables is critical to achieving the objective of modularity because it reduces the data demands at each step of the modeling process.

Throughout this paper we distinguish between component models, which have been presented thus far in a theoretical sense, and component model implementations. The component model implementations are the choices made by users of the PARRA framework about how a particular physical process will be represented, including what type of model to use (i.e., statistical vs. dynamical), the temporal and spatial resolution of analysis, etc. The state of atmospheric and hydrologic modeling is ever changing, and the "best" implementation choice depends on the modeler, the study area, and the intended end use (Baker et al., 2021). We have intentionally presented the PARRA framework in this section without tying the component models to specific implementations.

The PARRA framework has scientific value as an internally consistent and logically sound structure to connect atmospheric phenomena to community-level impacts. It enables the communication of ideas and results across disparate research fields, isolates the uncertainty associated with different processes within the model chain, and introduces new avenues for interdisciplinary collaboration. By contrast, implementing the PARRA framework for any given location imposes constraints, but also opens the door for practical insights. Site-specific implementations of the PARRA framework and component models are what quantify the probabilistic range of potential risk outcomes and generate actionable insights for stakeholders within case study communities.

## 3   Case Study: Sonoma County

We describe a proof-of-concept application of the PARRA framework for the lower Russian River in Sonoma County, California. Flood losses in Sonoma County have totaled over five billion dollars in the last forty years, with over 99% of that due to ARs (Corringham et al., 2019). Sonoma County also has the highest proportion of state disaster assistance payouts in California (34%), more than six times the second highest county (Sonoma County, 2017). Because of its location and its extensive history of AR-induced flooding (Ralph et al., 2006), Sonoma County is an excellent place to perform a case study demonstration of the PARRA framework. The case study is used to discuss implementation choices for each one of the component models introduced in the previous section and to illustrate the potential value and insights that can be provided by the PARRA framework.

The vast majority of AR-induced flooding in Sonoma County is the result of fluvial flooding from the lower Russian River, which has overtopped its banks 36 times in the last 80 years (Rogers, 2019). The Russian River flows 110 miles through

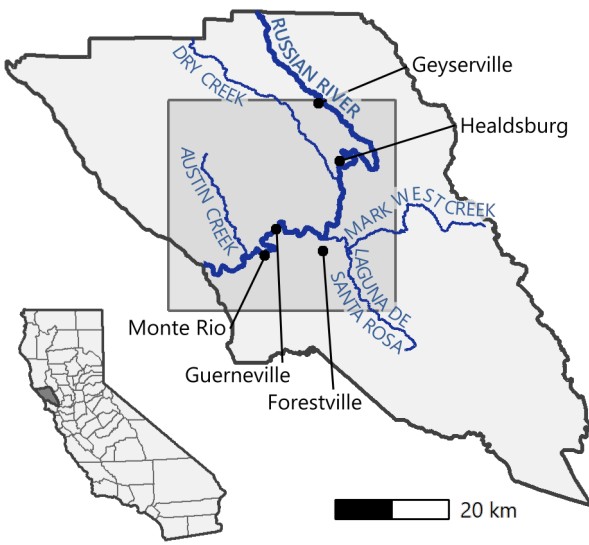

**Figure 2. Map of Sonoma County**. The shaded box indicates the study area, and the labels highlight notable hydrologic elements and riverside communities. A map of California with Sonoma County shaded in dark grey is included for geographic context.

Mendocino and Sonoma counties, draining a watershed that covers 1,485 square miles (3,846 km$^2$). The main tributaries within Sonoma County that flow into the Russian River are Dry Creek, which joins just south of Healdsburg; Mark West Creek, which joins around Forestville; and Austin Creek, which joins between Monte Rio and the Pacific Ocean. The Laguna de Santa Rosa is a protected wetlands complex that serves as an important overflow area for flood control. A map of the case study area and surrounding landmarks is included as Fig. 2.

Within Sonoma County we use the PARRA framework to examine the drivers and impacts of historical AR events. Each of the six subsections below corresponds with one of the pinch points defined in Sect. 2 and is divided into two parts. The first part of each subsection describes the user choices made to represent the study area within each component model. While we include many of the specific details related to fit and validation of these model implementations, the focus is on the overall workflow and how to functionally apply the PARRA framework. The second part of each subsection compares simulated Monte Carlo realizations of pinch point variables to observed data. These comparisons can be seen as a forensic reconstruction rather than an attempt to replicate the observed values. We focus on the new knowledge gained from the model implementations about how the observed values fall within the range of "what might have been."

We present two types of case studies to showcase the breadth and depth of insights that are possible in a model-by-model analysis. For breadth, we compare and contrast observed vs. simulated precipitation values for four different AR events. We examine storms with varying AR intensity categories to determine which storms displayed "average" behavior for their category and which exceeded predicted impacts. For depth, we focus the discussion for all other pinch points on a single Category 3 AR from February 2019, referred to as the February 2019 event. This event's recency combined with its severe impact mean

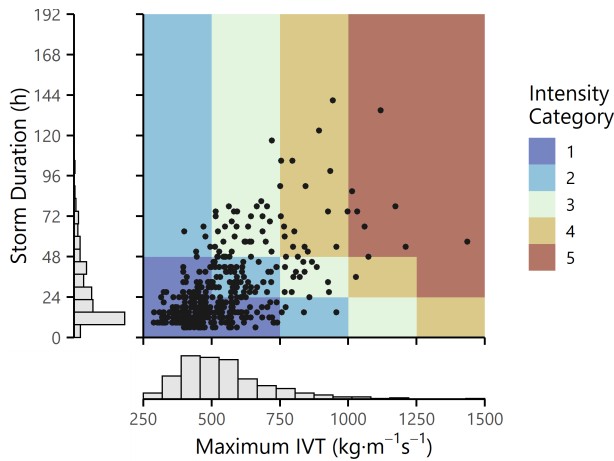

**Figure 3. Summary of Sonoma County historic catalog.** Each point represents one AR event recorded in Sonoma County between 1988 and 2019 ($n = 382$). The Ralph et al. (2019) intensity categories are represented by the background colors.

**Table 1.** Statistics of the Sonoma County historic catalog by AR intensity category.

| Category | Events | Mean Values | | | | |
|---|---|---|---|---|---|---|
| | | Maximum IVT $(\mathrm{kg \cdot m^{-1} s^{-1}})$ | Duration (h) | Precipitation (mm) | Peak Flow $(\mathrm{m^3 s^{-1}})$ | Time to Peak Flow (h) |
| 1 | 238 | 464 | 16 | 19 | 85 | 19 |
| 2 | 67 | 597 | 35 | 51 | 199 | 22 |
| 3 | 49 | 665 | 56 | 81 | 214 | 21 |
| 4 | 20 | 849 | 75 | 156 | 513 | 23 |
| 5 | 8 | 1,139 | 75 | 179 | 471 | 24 |

that datasets unique to this event are available to compare many of the individual component model implementations against ground-truth observations, allowing for a more focused analysis. Comparisons and results for additional storms can be found in the supplemental code release referenced at the end of the paper.

## 3.1 Atmospheric River (AR)

### 3.1.1 Component Model Implementation

The first step of the case study was to create a catalog of historic ARs affecting the study area. We started with IVT records from MERRA-2 (Gelaro et al., 2017), which records IVT at six hour intervals on a grid of $50\mathrm{km} \times 50\mathrm{km}$ cells. We then used the detection algorithm from Rutz et al. (2014) to identify landfalling ARs in each grid cell and recorded the maximum IVT

and duration for each. Because AR activity occurs almost exclusively during the first half of the water year (October $1^{st}$ –April $1^{st}$), we kept only ARs that occurred within those annual ranges.

A rectangular study area was defined as shown in Fig. 2 to encompass the lower Russian River, starting at USGS gage 11463500 near Geyserville and ending at the outlet to the Pacific Ocean. We followed the process outlined by Albano et al. (2020) to take ARs identified on the $50\text{km} \times 50\text{km}$ MERRA-2 grid and downscale them to the area of interest to create a catalog of historic AR events. The generated catalog contains 382 AR events recorded in the Russian River watershed over a 32 year period spanning water years 1988–2019. The maximum IVT and duration of these events are displayed in Fig. 3. The historic catalog is assumed to accurately represent the climatology of the region, although future work could expand upon this catalog to include the characteristics of storms not yet seen or recorded in Sonoma County.

We also collected additional information about each AR, including precipitation, streamflow, and more. Precipitation values are the storm-total cumulative precipitation areally averaged over the inlet watershed, and peak flow and time to peak flow were both calculated based on data from USGS gage 11463500 (study area inlet). Table 1 summarizes the statistics of some of these additional parameters as a function of the Ralph et al. (2019) AR intensity categories.

### 3.1.2 Case Study Events

Because we are focusing on case study events, we used real observed maximum IVT and duration values from the historic catalog as inputs to the precipitation model.

### 3.2 Precipitation (PRCP)

#### 3.2.1 Component Model Implementation

We used the historic catalog generated in the previous subsection to estimate a statistical relationship between the predictors (maximum IVT and duration) and the outcome (storm-total precipitation averaged over the watershed of interest). The goal in quantifying this relationship was to be able to generate simulated precipitation realizations that are consistent with the historic climatology of the region.

We implemented a weighted least squares (WLS) linear regression to predict precipitation as a function of maximum IVT, duration, and an interaction term between the two. We chose WLS over ordinary least squares methods because of the heteroskedasticity in the residuals, which can be identified by visual inspection of Figs. 4(a–b). Even after applying weights to correct for the heteroskedasticity there were still some extreme precipitation events in the historic catalog that were not well represented by a Gaussian error model. We therefore characterized the standard errors with a mixture model: 90% of residuals were calculated using the WLS standard errors, and 10% were calculated with a distribution fit the to the largest 10% of AR events (Bartolucci and Scaccia, 2005; Soffritti and Galimberti, 2011). Regression coefficients are reported in Eq. 3, and Figs. 4(a–b) show the regression line plotted over the historic catalog data at selected values of maximum IVT and duration.

$$\mathrm{E}[PRCP_i] = -3.52 + 0.0212 \cdot IVT_i - 0.504 \cdot DUR_i + (2.74\mathrm{e}{-3}) \cdot IVT_i \cdot DUR_i \tag{3}$$

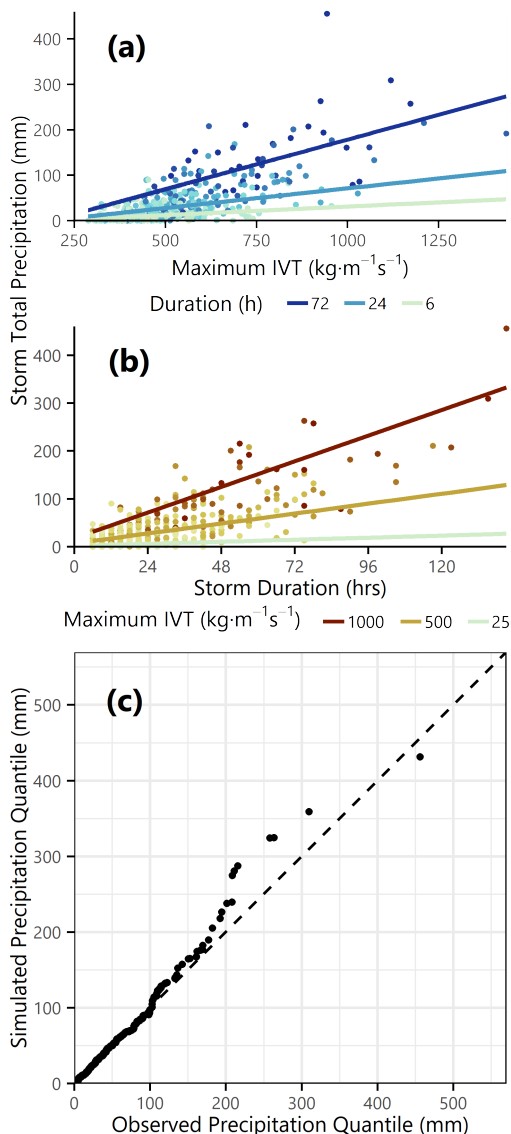

**Figure 4. Precipitation component model implementation.** Scatterplots of **(a)** maximum IVT vs. precipitation and **(b)** duration vs. precipitation, with the fitted WLS regression line shown for values specified in the respective legends. **(c)** Q–Q plot of observed vs. simulated precipitation including uncertainty for all AR events in the historic catalog.

where $\mathrm{E}[PRCP_i]$ is the expected total precipitation for event $i$, and $IVT_i$ (maximum IVT) and $DUR_i$ (duration) are the two elements of the pinch point variable $AR_i$.

Because our goal was to match the overall distribution of historical precipitation, we calculated goodness of fit metrics that focus on success in replicating distribution shape rather than individual events. Fig. 4(c) shows a quantile–quantile (Q–Q) plot

comparing a prediction from the fitted regression including errors against the observed distribution of precipitation from the historic catalog. Visually the simulated results fall very close to the parity line on this plot, which indicates good distributional fit. A more quantitative metric of distributional fit is the two sample Kolmogorov–Smirnov (K–S) test, which is a nonparametric test designed to measure the goodness of fit between two empirical distributions. We calculated the test statistic $d_{PRCP}$ such that a value of $d_{PRCP} > 1$ would reject with 95% confidence the null hypothesis that the two datasets are drawn from the same underlying probability distribution. Comparing the observed vs. simulated distribution yielded a test statistic of $\mathrm{E}[d_{PRCP}] = 0.478$, which is well under the rejection threshold for the null hypothesis.

### 3.2.2 Case Study Events

We present a comparison of observed vs. simulated precipitation values for four AR events. Figs. 5(a-b) are the two most recent Category 3 (strong) ARs in the historic catalog, and Figs. 5(c-d) are the two most recent Category 5 (exceptional) ARs. The dashed lines mark the recorded precipitation totals for each event and the tick marks along the top of the panel show the recorded totals from all ARs in the historic catalog in the same intensity category. For each event we generated 1,000 Monte Carlo realizations of precipitation given the observed maximum IVT and duration and plotted the resulting distribution as a histogram. The histograms represent realizations of potential precipitation if another AR occurred in Sonoma County with the same characteristics. We do not expect the observed dashed lines to fall in the center of the simulated distributions; rather, the observed values can be thought of as random samples from the simulated distributions, and comparing the two offers new insights into the character of specific ARs.

For example, Figs. 5(a) and 5(c) show two impactful storms for Sonoma County, from February 2019 and January 2017, respectively. The February 2019 event caused approximately $155 million in total damage (Chavez, 2019) and the January 2017 event caused approximately $15 million (County of Sonoma, 2017). While both events had precipitation totals in excess of 200 mm, the precipitation relative to the event-specific maximum IVT and duration was far higher in February 2019 than January 2017. The January 2017 event was a Category 5 AR, meaning that it had the greatest potential for hazardous impacts. Conditioned on the intense atmospheric conditions, though, the observed precipitation was near the mean of the simulated distribution in Fig. 5(c).

Figure 5(a) shows the predicted precipitation distribution for a Category 3 AR (mixture of beneficial and hazardous impacts) with the same maximum IVT and duration as was observed in February 2019. By all accounts, though, the February 2019 event was a very hazardous storm with severe impacts for communities in the study area. The observed precipitation is at the upper tail of what we would expect for a Category 3 event in both the observed distribution (shown in the tick marks at the top of the plot) and the simulated distribution (shown in the histogram). Therefore we infer that the precipitation is likely one of the drivers that led this particular AR to become a damaging event. In summary, the PARRA simulation results provide evidence that the January 2017 event was a moderate precipitation conditioned on extreme AR hazard while the February 2019 event was an extreme precipitation conditioned on more moderate AR hazard. These are two distinct pathways ARs can take to generate significant consequences.

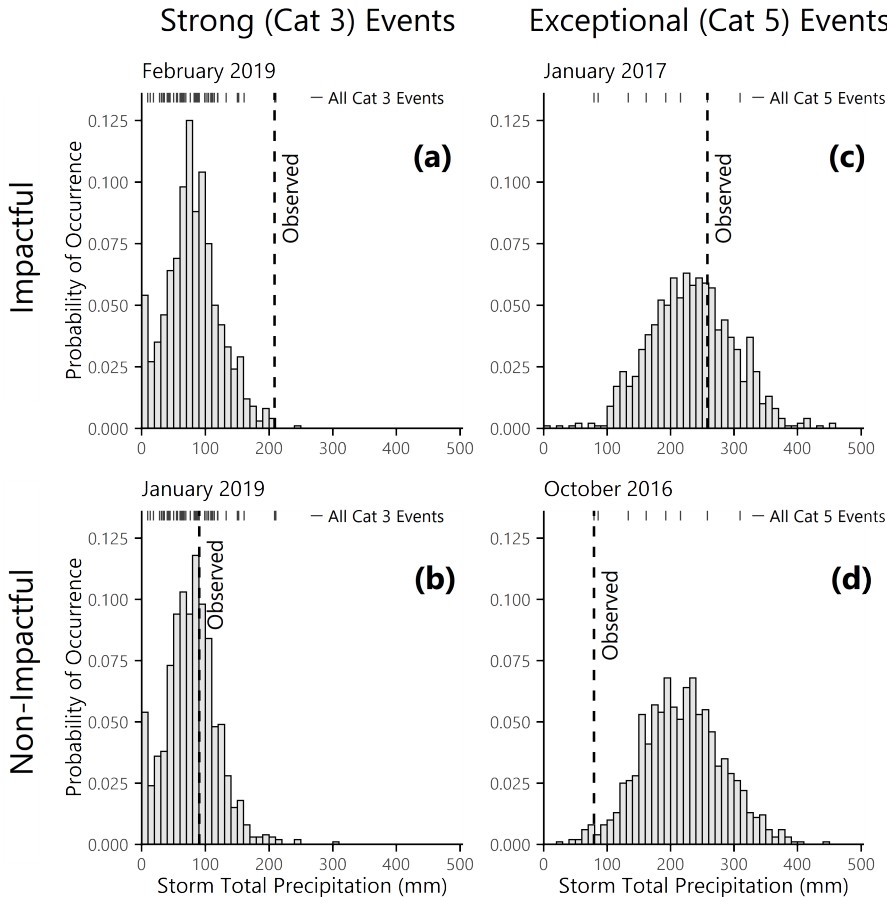

**Figure 5. Precipitation realizations for case study events.** Distribution of simulated precipitation realizations including uncertainty for AR events occurring in **(a)** February 2019, **(b)** January 2019, **(c)** January 2017, and **(d)** October 2016. Events are labelled by their Ralph et al. (2019) intensity category (left vs. right) and impact level (top vs. bottom). The observed precipitation for each event is marked by a dashed vertical line, and the tick marks along the top of each panel show how the observed values compare to precipitation totals from other AR events in the same intensity category.

We perform a similar comparison between Figs 5(b) (January 2019) and Fig. 5(d) (October 2016). Neither of these was an "impactful" storm: there were no state or federal disaster declarations, limited news coverage, and no reported loss totals. Both events had observed precipitation totals of about 90 mm, less than half the amounts seen in Figs. 5(a) and 5(c). The observed

precipitation total was in the middle of the simulated distribution for the Category 3 event in January 2017 but was on the low end for the Category 5 event in October 2016. The simulated results indicate that the AR event in October 2016 could have produced far more precipitation in the study area, and potentially far greater consequences, than what was actually realized. An interesting line of future research would be to examine these "near misses" to understand what factors drive certain events to produce extreme impacts and not others.

### 3.3 Hydrologic Conditions (HC)

#### 3.3.1 Component Model Implementation

To characterize antecedent hydrologic conditions we used the CPC Soil Moisture dataset as provided by the NOAA/ OAR/ESRL Physical Sciences Laboratory (van den Dool et al., 2003). This simulated dataset reports the monthly average soil moisture as an equivalent height of water (mm) found in the top meter of the subsurface. While this does not necessarily represent the true soil moisture in Sonoma County, the dataset has global coverage at a $0.5°$ latitude $\times$ $0.5°$ longitude resolution and reports monthly values back to 1948, which covers the full spatial and temporal extent of the historic catalog.

Soil moisture values were matched with AR events in the historic catalog, and based on these records the antecedent hydrologic conditions model was characterized as a lognormal distribution with $\mu_{\ln} = 4.18$ and $\sigma_{\ln} = 0.432$. When simulating future events we sample soil moisture from this distribution under the assumption that soil moisture records from past ARs implicitly capture the effects of seasonality on both AR occurrence and soil moisture. It is worth restating that the historic catalog generated for Sonoma County only considers ARs during the six-month wet season, so by construction only soil moisture values from October to April were included in this fitted distribution. The distribution of soil moisture values would shift downward if values from the dry season were included.

#### 3.3.2 February 2019 Case Study Event

We used the "observed" soil moisture for the February 2019 event (140 mm) as input for the streamflow component model, the next step of the model chain. This was a $96^{\text{th}}$ percentile soil moisture among all ARs in the historic catalog, indicating that the subsurface was already quite saturated in the study area when the AR made landfall.

### 3.4 Flow (Q)

#### 3.4.1 Component Model Implementation

As mentioned in Sect. 2.1, the streamflow hydrograph $Q$ is characterized in terms of three parameters: the peak streamflow ($Q_p$), the time to peak streamflow ($t_p$), and the hydrograph shape parameter ($m$). Equation 4 from the National Engineering Handbook (NRCS, 2004, Chapter 16) converts these three parameters into the full streamflow hydrograph for further analysis.

$$\frac{Q}{Q_p} = \left(\frac{t}{t_p}\right)^m \exp\left[m\left(1 - \frac{t}{t_p}\right)\right] \tag{4}$$

where $Q$ is the instantaneous streamflow timeseries recorded at USGS gage 11463500 (the study area inlet) at a vector of time values $t$, and $Q_p$, $t_p$, and $m$ are constant parameters defined for each AR event.

We step through the fit and calibration of each of these parameters starting with $Q_p$. Rather than using a complex hydrologic routing model, we implemented a simplified method to estimate $Q_p$ by calculating runoff as an intermediary variable. Runoff, the portion of precipitation that flows over the ground surface rather than contributing to evapotranspiration or infiltration,

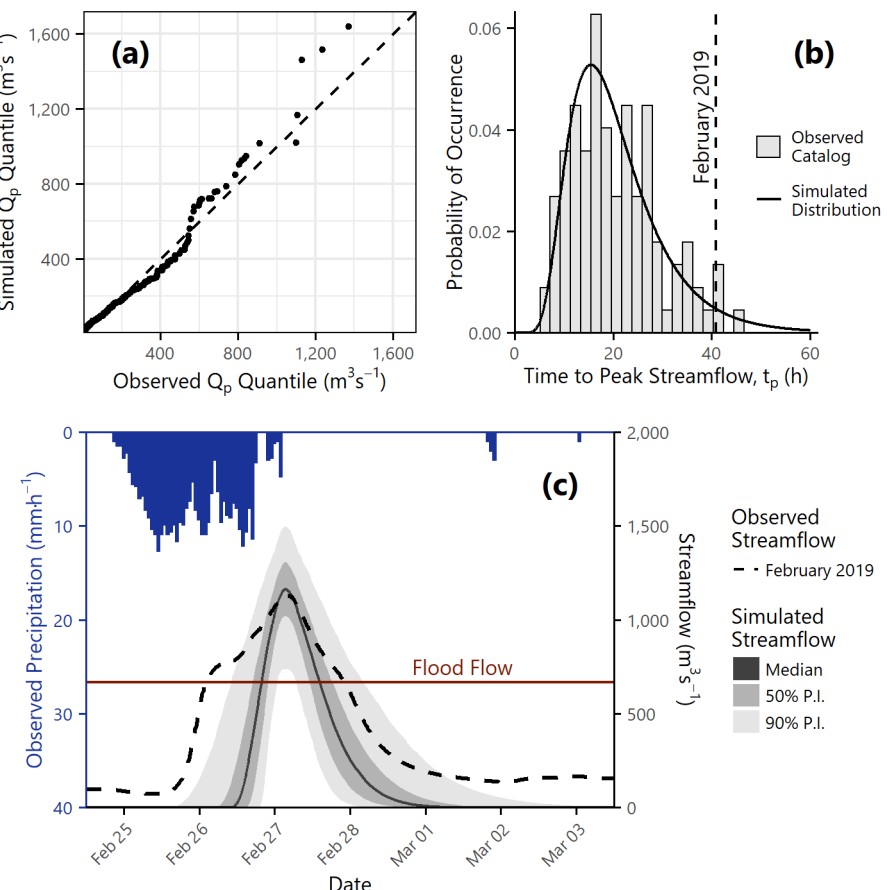

**Figure 6. Streamflow component model**. All values are calculated with respect to USGS gage 11463500 (study area inlet). **(a)** Q–Q plot of observed vs. simulated peak streamflow ($Q_p$) values for all events in the historic catalog. **(b)** Observed values from the historic catalog values vs. the fitted lognormal distribution for time to peak streamflow ($t_p$). The dashed line indicates the observed time to peak value for the February 2019 event. **(c)** Distribution of simulated streamflow hydrograph realizations for the February 2019 event. The left axis represents observed hourly precipitation and the right axis represents streamflow. The observed hydrograph timeseries is shown as a black dashed line. The solid line represents the median of the simulated realizations, and the dark and light grey shaded areas represent the 50th and 90th percentile prediction intervals, respectively. The dark red horizontal line indicates the National Weather Service (NWS) flood flow for USGS gage 11463500 (39.7 ft or 665 $\mathrm{m^3s^{-1}}$).

was calculated for each event in the historic catalog using the empirical curve number method (NRCS, 2004, Chapter 10). An
ordinary least squares linear regression was then fit to the historic catalog to estimate $Q_p$ as a function of precipitation, runoff, and an interaction term between the two. We used the same mixture model introduced in Sect. 3.2 to capture the long tails of the residual distribution, with 90% of errors calculated based on the bulk of the historic catalog values and 10% calculated based on the extremes. The regression form is shown in Eq. 5, and the Q–Q plot for this regression fit is shown in Fig. 6(a).

We again validated regression fit by comparing the shape of the observed vs. simulated distribution rather than comparing individual records. The K–S statistic for the OLS regression fit was calculated to be $\mathrm{E}[d_Q] = 0.889 > 1$, so we conclude that the regression produced an acceptable fit to the data with 95% confidence.

$$\mathrm{E}[Q_{p,i}] = 8.99 + 0.363 \cdot PRCP_i + 11.6 \cdot R_i - 0.0162 \cdot PRCP_i \cdot R_i \tag{5}$$

where $\mathrm{E}[Q_{p,i}]$ is the expected peak streamflow at USGS gauge 11463500 during event $i$, $R_i$ is the watershed-average runoff, and $PRCP_i$ is the watershed-average total precipitation.

The time to peak streamflow $t_p$ was calculated based on the distribution of observed values in the historic catalog, which was found to be well represented by a lognormal distribution with $\mu_{\ln} = 2.94$ and $\sigma_{\ln} = 0.443$. This distribution is shown in Fig. 6(b).

For the hydrograph shape parameter we chose $m = 4.0$, which was recommended by the National Engineering Handbook (NRCS, 2004, Chapter 16) and was found to be a reasonable approximation for this section of the Russian River through comparison to observed streamflow records.

### 3.4.2 February 2019 Case Study Event

The hydrograph at USGS gage 11463500 (study area inlet) recorded a peak streamflow of $Q_p = 1{,}130 \, \mathrm{m^3 s^{-1}}$ and a time to peak streamflow of $t_p = 41$ hrs. Given the February 2019 observed precipitation and antecedent soil moisture, we generated 1,000 Monte Carlo realizations from the streamflow model and compared the predicted streamflow hydrograph from the calibrated component model implementation to the observed hydrograph from the February 2019 event. Using observed data as input rather than the simulated distributions from Sects. 3.2 and 3.3 allows us to isolate the uncertainty associated with this specific step of the model chain in isolation.

The complex shape of the observed streamflow timeseries in Fig. 6(b) is a function of the unique watershed response as well as the spatial and temporal heterogeneity of the input precipitation. By contrast, the simulated distribution is based on the unit hydrograph method, which assumes that the precipitation distribution is uniform and that all runoff enters the channel at a single location. This limits our ability to capture certain kinds of behavior, such as the early peak seen in the observed streamflow timeseries in Fig. 6(c). The early peak could be due to catchment processes that cause a lagged tributary response, input from direct surface runoff, spatial variation in precipitation intensity and duration, or any number of other mechanisms. We include the observed hyetograph at the top of the plot in Fig. 6(b) to show just one aspect of the natural variability that affects the observed timeseries.

Despite the simplification imposed by the unit hydrograph method, many metrics of interest are reasonably well characterized by the simulated timeseries. The observed peak streamflow ($1{,}130 \, \mathrm{m^3 s^{-1}}$) is at the $43^{\mathrm{rd}}$ percentile and the observed floodwave duration (81 h) is at the $67^{\mathrm{th}}$ percentile of the respective simulated distributions. Recall from Sect. 3.2 that the observed precipitation was notable high conditioned on the observed atmospheric conditions. We now note that while the observed streamflow may have been high for a Category 3 event, it was in the middle of the simulated distribution conditioned

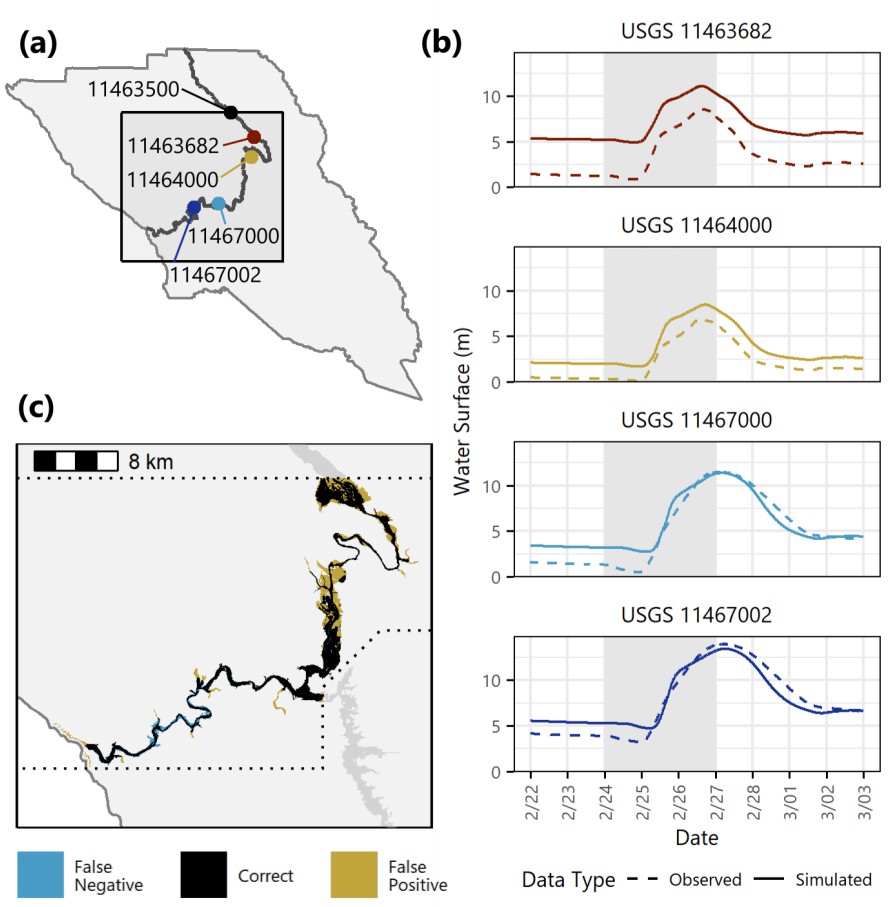

**Figure 7. Inundation component model.** **(a)** Locations of USGS gages with available stage data for the February 2019 event. **(b)** Comparison of observed vs. simulated streamflow at selected USGS gage locations for the February 2019 event. The Y axis represents the water surface height above the datum of the gage, as recorded in the USGS National Water Information System. The grey shaded areas indicate the days with sustained AR conditions. **(c)** Comparison of "observed" (Sonoma GIS) vs. simulated inundation extent for the February 2019 event. The dotted lines denote the extent of the Sonoma GIS map, which is slightly smaller than the PARRA study area and does not include the Laguna de Santa Rosa.

on the observed precipitation. Therefore we conclude that the hydrologic routing was likely not one of the physical processes contributing to the "extremeness" of the February 2019 event.

### 3.5 Inundation (INUN)

#### 3.5.1 Component Model Implementation

The inundation model accepts a hydrograph at the inlet to the study domain, routes water through the river channel, and distributes it through the floodplain based on site-specific information defined by the user. We used the hydrodynamic model LISFLOOD-FP (Bates and De Roo, 2000), referred to as LISFLOOD, to perform these calculations as it is lightweight and computationally efficient yet capable of capturing hydrologic processes over complex terrain. The model grid was defined with 925,000 cells at $40\text{m}\times40\text{m}$ resolution. LISFLOOD has several parameters, including floodplain roughness, channel roughness,

channel shape, and hydrologic boundary conditions, that must be specified by the user to best represent the study area. We chose twenty parameters of interest within the LISFLOOD environment, generated 500 Latin hypercube sample sets of these twenty parameters, and calculated inundation maps for each sample set. We then conducted sensitivity testing to determine best-fit parameters for the subset of "sensitive" parameters that were determined to significantly impact inundation results. Best-fit values were chosen such that when the 100-year peak flow value from USGS StreamStats was given as input, the LISFLOOD

inundation extent matched the 100-year floodplain from the FEMA National Flood Hazard Layer (NFHL) along the Russian River. We adopted the accuracy metrics from Wing et al. (2017) used for validation of their nationwide flood hazard map and referred to their values as benchmarks of acceptable performance. Overall the fitted LISFLOOD model was able to reach a critical success index of 69%, which means that when either the FEMA NFHL or the LISFLOOD model predicted inundation, the prediction of the LISFLOOD model was correct over two thirds of the time.

Despite its efficiency, the runtime of LISFLOOD was found to be prohibitive for Monte Carlo analysis, which involves repeated iterations of the same calculations. Therefore we calibrated and used a low dimensional surrogate model to quickly and accurately reproduce the results of the hydrodynamic simulation. Surrogate models are popular for testing multiple scenarios, exploring uncertainty, and making near-real-time predictions without the time and computational expense of high-fidelity model runs (Bass and Bedient, 2018; Razavi et al., 2012). 1,000 LISFLOOD runs were conducted over a range of $Q_p$ and $t_p$

values to populate the parameter space. We used the inverse distance weighting spatial interpolation method to generate new inundation maps based on this existing database of LISFLOOD runs. Based on a specified $Q_p$ and $t_p$, the spatial interpolator identifies the "closest" points in parameter space and weights them based on distance to produce a best-fit estimate of the LISFLOOD inundation map. The hyperparameters of the surrogate model, which control the size of the search neighborhood, the distance weighting power function, and the relative importance of $Q_p$ vs. $t_p$, were fit by ten-fold cross-validation. The

error metric was the root mean squared error (RMSE) of all LISFLOOD grid cells. Replacing the LISFLOOD hydrodynamic simulation with this new surrogate model significantly reduced the computational demand of the PARRA framework while maintaining high levels of accuracy. The final fitted surrogate model reduced the runtime of a single inundation calculation from hours to seconds and had a median RMSE of 3.5 cm, which is a tolerable tradeoff when compared to the 3 cm median relative vertical accuracy reported by the digital elevation model. Additional information, including data, documentation,

and reproducible code, to replicate the fit and calibration of both LISFLOOD and the surrogate model can be found in the supplemental code release, which is referenced in the "Data and code availability" section at the end of this paper.

The gridded inundation maps from the surrogate model were overlaid with building information to estimate inundation heights at locations of interest, namely residential buildings in Sonoma County. We used building footprints from 2019 SonomaVegMap LIDAR data and building parcel information from the 2021 Sonoma County Clerk Recorder Assessor to identify residential locations within the study area and estimated a count of about 41,000 homes.

### 3.5.2 February 2019 Case Study Event

Given the February 2019 observed hydrograph at USGS gage 11463500 (study area inlet), we used LISFLOOD to generate a simulated inundation map for the February 2019 event. Because the N-dimensional inundation pinch point variable $INUN$ contains significantly more data than we have generated thus far, we explore three strategies to compare the observed and simulated inundation maps. The first focuses on the height of the river within the channel. There were four USGS gages downstream of the study area inlet that recorded stage heights at 15 minute intervals during the February 2019 event, as shown in Fig. 7(a). Each of the timeseries plots in Fig. 7(b) compares the observed vs. simulated stage height in the channel at one of these four locations.

Outside of the channel there exists little recorded information about floodplain inundation resulting from the February 2019 event. However, the Geographical Information Systems (GIS) branch of the Permit and Resource Management Department in Sonoma County has made available some of the results from their internal inundation modeling of the lower Russian River. These maps, rather than representing specific scenario flood events, are indexed to stage heights at the Guerneville bridge (USGS gage 11467002, Fig. 7(a)). Together they constitute a suite of design events that model stepwise increases in inundation severity across the Russian River watershed. We chose the inundation map from the design event that most closely matched the peak stage height observed at USGS gage 11467002 during the February 2019 event, referred to as "the Sonoma GIS map." While we cannot expect a perfect match, as there are specifics of hydrograph shape and site conditions from the February 2019 event that are not captured in an idealized design event, we assume that the Sonoma GIS map serves as a reasonable representation of the true inundation observed during this AR. Our LISFLOOD model was able to reproduce the Sonoma GIS map with a critical success index of 67.77%, which indicates that the observed inundation is within the range of what we would reasonably predict given the observed hydrograph. Figure 7(c) shows the spatial extent of flooding from both sources.

As a third and final comparison strategy we calculated the number of inundated buildings due to both the Sonoma GIS map and the LISFLOOD model as a proxy for flood impacts. News reports following the February 2019 event estimated that about 1,900 homes were damaged (Chavez, 2019). When overlaid with residual building locations, the Sonoma GIS map estimated 1,678 buildings with nonzero flood height, and the LISFLOOD model estimated 1,380.

## 3.6 Damage Measure (DM)

### 3.6.1 Component Model Implementation

The damage measure ($DM$) is the ratio of expected building repair costs to building value and is computed from inundation using a two step process. First, the first floor water level at each building is calculated by taking the inundation height relative

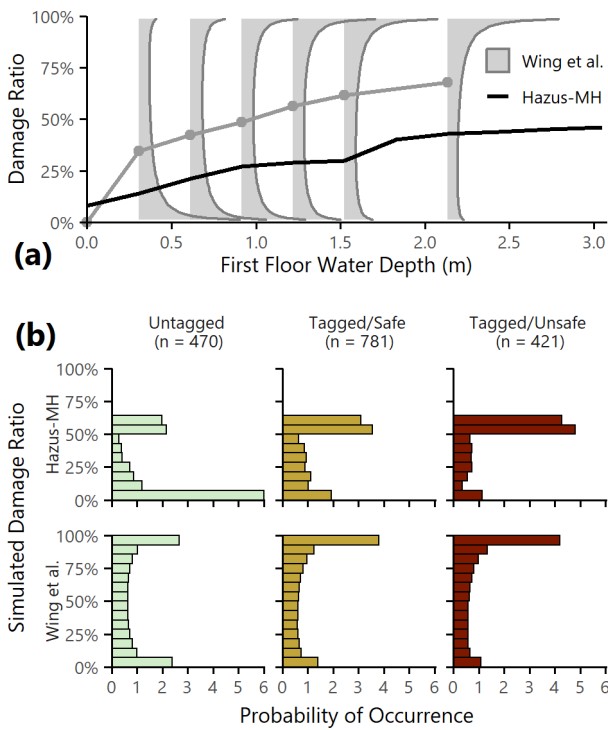

**Figure 8. Damage component model. (a)** Depth-damage relationships from Wing et al. (2020) and from Hazus-MH. The light grey areas represent the shapes of the beta distributions for the five defined curves (1, 2, 3, 4, 5, & 7 ft), and the grey line follows the distribution means. The black line is the Hazus-MH depth-damage function for single story residential buildings without a basement. **(b)** Damage ratio distributions for inundated buildings based on 1,000 Monte Carlo realization, using the Hazus-MH (top) and Wing et al. (2020) (bottom) depth-damage relationships and grouped by safety category.

to the ground surface ($INUN$ from the previous component model) and subtracting the height of the foundation. Foundation

height information was collected at an aggregate level from Hazus-MH (FEMA, 2020), then assigned to individual buildings based on the distribution of foundation types and heights within that building's census tract. The foundation options were: basement (6 ft below grade), slab (1 ft above grade), crawl space (3–4 ft above grade), piers (5–6 ft above grade), or piles (7–8 ft above grade).

     Second, the first floor water level was converted to a damage ratio using a depth-damage curve. We chose two sets of

relationships: Hazus-MH depth-damage curves, which are widely used in engineering applications, and the beta distributions from Wing et al. (2020), which capture the uncertain and bimodal nature of residential flood damage at the household scale. A comparison of the damage ratios predicted by these relationships at various water depths is provided in Fig. 8(a).

**Table 2.** RESA tags and inundated buildings in the study area following the February 2019 event.

|  | Untagged | Tagged/ Safe | Tagged/ Unsafe |
|---|---|---|---|
| Number of Buildings | 10,513 | 820 | 433 |
| Inundated Buildings | 469 | 788 | 421 |
| Percent Inundated | 4.46% | 96.10% | 97.23% |

### 3.6.2   February 2019 Case Study Event

Because there was little available in the way of site-specific damage information, we used building safety as a proxy variable to
facilitate investigation of observed vs. predicted damage. Sonoma County performed over 2,000 building inspections as part of
the Rapid Evaluation Safety Assessment (RESA) that immediately followed the February 2019 event. Buildings were assigned
colored tags based on these rapid inspections: green tags indicated that the structure was safe to enter, yellow indicated some
risk, and red indicated an imminent safety threat. The RESA tags are categorical measures of safety and are thus an imperfect
analog to the continuous damage ratios estimated by the damage model. A building may have been tagged as unsafe for reasons
beyond just inundation (i.e., roof damage or downed trees), and conversely a building that experienced nonzero inundation
may still have been deemed safe to enter. Despite these limitations, comparing tagged buildings to our prediction of damaged
buildings provides some intuition that damage is being predicted where actual damage was likely to have occurred.

   We matched 1,253 of the RESA tags to residential buildings along the Russian River and aggregated them into three safety
categories: untagged, tagged/safe (green), and tagged/unsafe (yellow+red). Using the Sonoma GIS map from the previous sub-
section we estimated how many buildings within the study area saw nonzero inundation at the ground surface (not accounting
for foundation height). We see in Table 2 that 96–97% of all tagged building locations were predicted to have some level of
inundation, compared to less than 5% for untagged building locations. This is consistent with our expectations, because by
definition any building with a RESA tag was one that was identified as a top priority for safety inspection and was therefore
much more likely to have experienced inundation compared to an untagged building.
Figure 8(b) shows the expected distributions of damage ratios by safety category among the inundated buildings in the study
area, as a function of the Hazus-MH (top) and Wing et al. (2020) (bottom) damage relationships. Although the shapes of the
damage ratio distributions are notably different, we see overall increasing levels of damage with increasing tag severity. Of the
small proportion of untagged buildings that saw nonzero inundation, most were predicted to receive little to no damage, and

the tagged/unsafe category saw more severe damage outcomes than the tagged/safe category. These plots show that we are, in a broad sense, capturing more intense damage where we expect to do so.

## 3.7 Decision Variable (DV)

### 3.7.1 Component Model Implementation

The final component model in the PARRA framework converts the building-level damage ratios to a decision variable. The decision variable ($DV$) is some measurement of the impact or consequence of a hazard event. The consequences of flooding are traditionally described using the two axes of direct vs. indirect and tangible vs. intangible (Merz et al., 2004). This creates four categories: direct tangible (e.g. structural damage), indirect tangible (e.g. business interruption costs), direct intangible (e.g. loss of life), and indirect intangible (e.g. post-traumatic stress). The decision variable of interest for this case study is direct tangible loss, specifically the estimated total repair cost of residential buildings.

Losses were estimated for each building by multiplying the damage ratio times the expected value of the building as determined from Sonoma tax assessor roll data. Tax assessor data has some inherent limitations in California due to Proposition 13, which in many cases prevents assessments from directly tracking property values, so we applied correction factors at a census tract level such that the median from the tax assessor roll matched the median value of owner-occupied housing units reported by the American Community Survey (ACS). When used in conjunction with the ACS correction factors, the high-resolution tax assessor dataset allows us to match valuations to inundated buildings at the household level and more precisely estimate resulting losses.

### 3.7.2 February 2019 Case Study Event

It was not possible to validate the loss model individually as we have done for all preceding component model implementations. With neither damage ratios nor loss information at the household level, we had no accurate input data and no observed response data to compare against the simulated output. Instead we present a fully probabilistic estimate from the entire PARRA framework in the next section.

## 4 Results

In this section we utilize the framework to move beyond individual component models to capture the broad spectrum of potential AR-driven fluvial flood losses in Sonoma County, starting with scenario events and moving to a characterization of the full distribution of AR flood risk. We define and report metrics of interest for community loss assessment, then consider how the framework can be used to evaluate potential mitigation policies.

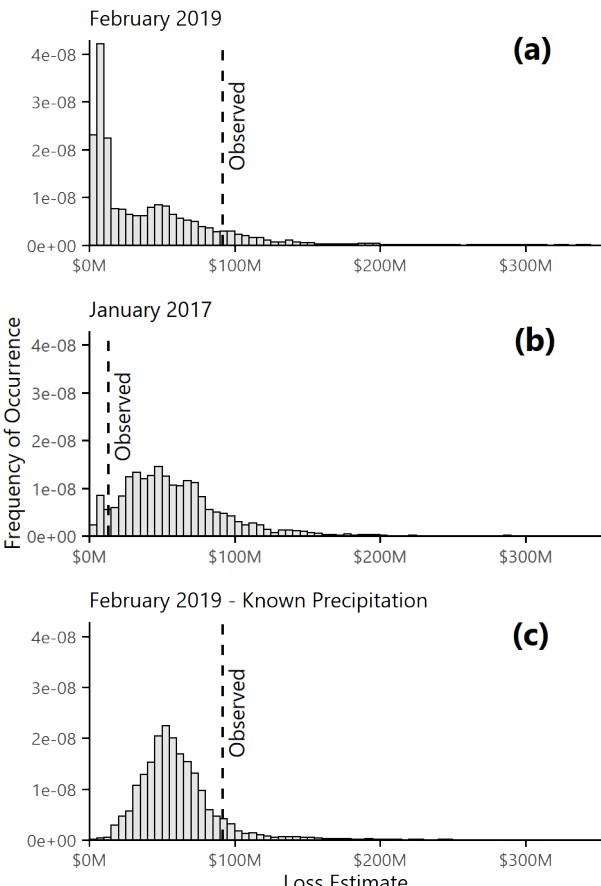

**Figure 9. PARRA simulated loss results.** Observed vs. simulated losses for **(a)** the Category 3 February 2019 event, **(b)** the Category 5 January 2017 event, and **(c)** a modified version of the February 2019 event with perfect information about expected precipitation. In each plot the dashed vertical line marks the observed loss estimate.

## 4.1 Scenario Events

We consider loss distributions for the Category 3 AR from February 2019 and the Category 5 AR from January 2017, both introduced in Sect. 3.2. Given the observed maximum IVT and duration values and the "observed" soil moisture values for our scenario events, we ran all of the component models in sequence and generated 10,000 probabilistic loss realizations to estimate the distribution of potential loss outcomes. These are the flood losses that *could have occurred* for each event if any realizations of the other pinch point variables had been different; i.e., if the precipitation total had been lower (see Fig. 5), if the streamflow peak had lasted longer (see Fig. 6), etc. We compare the observed vs. simulated losses and examine how the losses were spatially distributed within the study area.

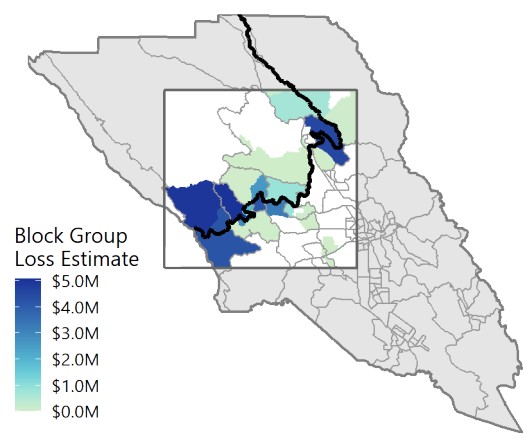

**Figure 10. Spatial distribution of PARRA losses.** Spatial distribution of mean losses due to the February 2019 event, aggregated to the census block group level.

The histogram of simulated loss realizations for the February 2019 event is shown in Fig. 9(a). The observed maximum IVT was $620 \text{ kg} \cdot \text{m}^{-1}\text{s}^{-1}$ and the observed duration was 57 h. The vertical dashed line marks \$91.6 million, the estimate of true losses experienced by residential buildings in Sonoma County (Chavez, 2019). The PARRA framework estimates this historical event to have been an 89[th] percentile loss event based on the driving AR characteristics. We have previously stated that the February 2019 event was a moderate storm in atmospheric terms that generated severe hydrologic and economic effects. There is significant bias at work in the selection of this case study event, because by definition the noteworthy events in the historic catalog are those with the highest impacts. If we understand the true loss to be only one stochastic realization out of the set of all possible losses that could have occurred due to the February 2019 initial conditions, and we consider that the event was selected because of the severity of its observed impacts, it is reasonable that the observed loss estimate comes from the upper tail of the simulated distribution. Note also that the while the loss for the February 2019 event was higher than expected for its AR characteristics and antecedent conditions, approximately 10% of simulations produced even more extreme losses. Our results indicate that the observed loss of \$91.6 million is not necessarily the worst-case scenario of what we could have seen.

Figure 9(b) shows the simulated and observed loss results for the Category 5 AR occurring in January 2017. This event had a maximum IVT of $1{,}173 \text{ kg} \cdot \text{m}^{-1}\text{s}^{-1}$ and a duration of 78 h, much larger than February 2019. This AR was one of the first major precipitation events in the 2017 water year, which came after multiple years of drought conditions in northern California. The observed loss thus falls at the low end (7[th] percentile) of what was expected for an AR of this magnitude. The January 2017 event was also the first in a series of strong to exceptional ARs that lasted about six weeks and led to severe statewide consequences, notably a damaging overflow event at the Anderson Dam in San Jose and a spillway failure at the Oroville Dam that led to emergency evacuation of almost 200,000 people. The 2017 AR sequence underscores the importance of initial conditions in the modeling of extreme events in northern California. While the PARRA framework captures initial

soil moisture conditions, it does not currently capture sequential and compounding events. This could be included in future
implementations of the PARRA framework and is an interesting potential avenue for future exploration.

Because of the probabilistic nature of the PARRA framework, its strength lies not in the reproduction of specific past events, but in quantifying total risk and assessing the relative differences between alternative decision pathways. The results in Fig. 9(a) assume that no information is known about the storm other than the maximum IVT, duration, and soil moisture. However, AR forecasts now typically include an estimate of the expected regional precipitation total. If we had perfect information about

560 total precipitation (i.e., we could predict in advance exactly what the observed value would be) we could start the PARRA framework at the precipitation pinch point $PRCP$ and run all subsequent component models in the sequence probabilistically. Figure 9(c) is therefore an exploration of a "what-if" scenario where losses are conditional on the observed precipitation value from February 2019 rather than the AR characteristics. While the observed \$91.6 million loss estimate is similarly extreme in this case (91$^{st}$ percentile event vs. 89$^{th}$) and the tail behavior of the two distributions is about the same, the body of the

565 distribution in Fig. 9(c) shifts to larger losses, and the probability of seeing zero-loss events nearly disappears. Calculating the differences between the loss distributions conditioned on different sets of input information can serve to quantify the value of more accurate AR forecasting tools for the study area.

Figure 10 shows the spatial distribution of building losses from the February 2019 event, averaged across all Monte Carlo realizations and aggregated to the census block group level. Losses are concentrated along the banks of the Russian River with

570 hotspots near Healdsburg, Guerneville, and the mouth of the river near the Pacific Ocean. These locations received warnings and evacuation orders before and during the storm event. While these particular communities are already known to have high vulnerability to flooding, the PARRA simulation results offer a new way to quantitatively prioritize investments in flood mitigation, from emergency communications to infrastructure projects to high-resolution modeling.

## 4.2 Average Annual Loss

To represent the full spectrum of AR hazards we move from examining case study events to analyzing the whole historic catalog. The first metric of interest for summarizing and communicating probabilistic risk is the average annual loss (AAL), or the long-run average of expected loss per year. The AAL is widely used in the insurance sector to price policies and is a convenient and well-known metric to summarize risk.

We generated 38,200 simulated loss realizations in our study area: 382 events in the 32-year historic catalog times 100

realizations per event. This produced a synthetic stochastic record representing 3,200 ($32 \times 100$) years of potential AR losses. To calculate the AAL we rank-ordered the stochastic record, assigned each event a $\frac{1}{3200}$ annual rate of occurrence, and summed the product of event rate times expected losses as described in Eq. 6. This calculation produces an empirical estimate of the AAL due to AR-induced flooding (Grossi and Kunreuther, 2005; Baker et al., 2021).

$$\mathrm{E}[DV] = \sum_{i=1}^{38,200} dv_i \cdot \lambda(event_i) \tag{6}$$

where $event_i$ is one event in the stochastic record, $dv_i$ is a sample realization of the decision variable $DV$ representing the expected losses due to event $i$, and $\lambda(event_i) = \frac{1}{3200}$ is the annual rate of occurrence.

The mean AAL estimated from the stochastic record for AR-induced flood losses to residential structures is $111 million, with 90% confidence that it lies between $92−$132 million. We compared this to another loss estimate based on claims from the National Flood Insurance Program. We collected flood insurance claims within the census tracts that overlap with the study area for the years 1979-2018, then estimated the flood insurance penetration rate in these census tracts by dividing the average annual number of policies by the total number of households as determined from the American Community Survey. The average annual loss for the study area is then the average annual insured loss divided by the insurance penetration rate, which is found to be $121 million. It is important to acknowledge the significant uncertainty around both of these estimates that stems from the difficulties in characterizing the types of low probability, high consequence events that cause damaging impacts. However, we can use this comparison to confirm that the PARRA framework is able to broadly capture the magnitude of AR-induced losses along the lower Russian River.

## 4.3 Loss Exceedance Curve

The second metric is the loss exceedance curve, which measures the expected annual rate of occurrence for a range of possible losses. The loss exceedance curve was introduced in Sect. 2 as the final outcome of the PARRA framework. This curve is the desired end product of a probabilistic risk assessment, summarizing information about expected likelihood and consequence across a range of potential risk events. It also provides insight about the character of the risk, such as whether frequent small events or a few rare events dominate the AAL. To our knowledge there has never been a loss exceedance curve created for AR-induced flood risk in Sonoma County.

We use the synthetic stochastic record generated above as a modeled representation of the event space. The loss exceedance curve is estimated by evaluating Eq. 7 at a range of $x$ values. The result is shown as the black line in Fig. 11.

$$\lambda(DV > x) = \sum_{i=1}^{38,200} \mathrm{I}(dv_i > x) \cdot \lambda(event_i) \tag{7}$$

where $\mathrm{I}(dv_i > x)$ is the binary indicator function measuring whether the the loss for event $i$ exceeds the specified threshold $x$. Equation 7 is the empirical approximation of Eq. 1, using sample realizations of $dv_i$ in place of an analytical $P(DV > x \,|\, AR)$ Baker et al. (2021).

With the loss exceedance curve we can examine risk thresholds such as the "100-year event" (1% annual rate of occurrence). This event, marked by the horizontal dashed line in Fig. 11, has a loss of $225 million. Figure 11 also indicates that the loss associated with the February 2019 event, $91.6 million, has an annual occurrence rate of $\lambda = 0.095$ (approximately a 1-in-11 year event).

## 4.4 Mitigation Analysis

A benefit of taking a performance-based approach is the ability to set a specified performance objective, such as loss reduction, and determine what changes can be made to the hazard, exposure, and vulnerability to reach that target. Working backwards

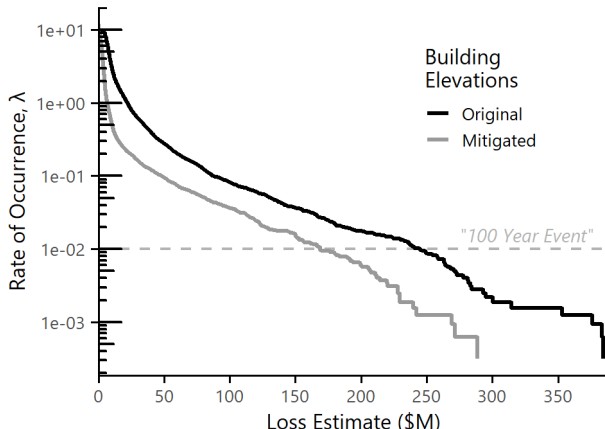

**Figure 11. AR flood loss exceedance curve for Sonoma County.** Loss exceedance is shown before and after the hypothetical mitigation action of elevating 200 households to reduce the AAL by half.

to design a system that meets a set performance target is a powerful and unique capability of performance-based frameworks. Here we demonstrate the performance-based aspect of the PARRA framework through a hypothetical mitigation analysis. We define a target loss reduction threshold of reducing the AAL by half, and we assess the effectiveness of home elevation as a
pathway to meet that threshold. We then quantify the effects of the system changes on the shape of the loss exceedance curve to highlight the framework's capability to prospectively assess events without historical precedent.

Sonoma County is well aware of its flood risk and since 1995 has spent almost $60 million in 2019-adjusted dollars through the FEMA Hazard Mitigation Assistance Program to acquire or elevate 542 private properties. We explored the benefit of a hypothetical extension of the home elevation program and estimated how many homes would need to be elevated in order to
625  reduce the AAL by half, from $111 million to $55 million. We ranked homes in order of increasing distance from the Russian River, elevated them one by one, and iteratively calculated the new AAL until it fell below the target threshold. We found that this strategy would require elevating approximately 200 additional homes in order to achieve the performance target. This is not necessarily the most efficient or effective possible strategy, but it illustrates in a straightforward manner how prospective evaluations can be performed.

630  The resultant change between the original and mitigated loss exceedance curves is shown in Fig. 11. The horizontal distance between the original and mitigated curves represents the benefit of mitigation at a specified rate of occurrence $\lambda$. For the most frequent events ($\lambda > 1$) this distance is relatively small, but the benefit increases for larger events. The 100-year loss estimate drops from $225 million to $169 million, a difference of over $50 million dollars. The expected benefit of mitigation increases for events with smaller rates of occurrence, reaching over $75 million for the largest events in the record.

635  In addition to understanding the difference in expected loss at a constant return period, we can also pick a constant loss value and examine how the return period of that event has changed. For example, the February 2019 case study event had a flood loss of $91.6 million, which was originally a 1-in-11 year loss. The new estimated occurrence rate for an event similar to

the February 2019 case study is $\lambda = 0.0413$, or a return period of approximately 24 years. This analysis indicates that there is significant potential benefit to continuing investment in home elevations as a flood loss mitigation strategy in Sonoma County. These types of insights can help community members and stakeholders make more informed decisions about their level of flood risk and the effect of potential mitigation actions.

## 5 Discussion

Implementing a large, complex, multi-disciplinary model sequence such as the PARRA framework involves challenges and compromises, but offers significant potential for novel insights. In this section we highlight the more subtle benefits of a probabilistic, performance-based approach and discuss some of the practicalities involved in implementation and validation.

### 5.1 Framework Implementation

The PARRA framework represents one end of the continuum between stochastic and deterministic modeling. It will not perform as well as a high-fidelity multi-scale physics model calibrated to a given set of input forcings for a specific scenario event, and it is not intended to replace existing models designed for that use case. However, it is impossible to scale the granular analysis performed by deterministic models to produce a probabilistic estimate of flood risk across a range of potential AR events (Apel et al., 2004; Savage et al., 2016). The PARRA framework thus serves a fundamentally different purpose within the literature of risk—if deterministic modeling gives us the best possible representation of a single tree, then the PARRA framework aims to characterize the shape and scale of the entire forest.

A key functionality of the PARRA framework is its ability to track and quantify uncertainty across multiple component models. Each component model implementation produces a best-fit estimate and a range of uncertainty for the output pinch point variable of interest through Monte Carlo simulation. The benefit of Monte Carlo simulation is that the component models do not need to be resolved into analytical PDFs; instead pinch point variables are calculated directly using the component models, then propagated through the full equation to produce an empirical distribution of total loss. However, a downside is the computational effort involved in generating these realizations, especially for time- or resource-intensive component models.

We came up with several practical solutions to minimize the computational expense of the PARRA framework and bring it into the range of procedural feasibility without compromising accuracy. Two of those solutions are highlighted here. First, we implemented a functional programming paradigm in R to run the framework and leveraged parallelization and high performance computing resources to speed calculations. Second, we limited computational costs of individual component models by relying on statistical relationships rather than high-fidelity physical models (e.g., $f(Q|PRCP,HC)$, Sect. 3.4) and implementing surrogate models to replace expensive calculation steps (e.g., $f(INUN|Q)$, Sect. 3.5). The repetitive nature of Monte Carlo simulation means that savings in any one component model lead to multiplicative effects when running the entire framework in sequence.

## 5.2 Validation Data

The PARRA framework is a risk analysis tool that is globally applicable and can be used to assess AR flood risk at any scale. However, the implementation of the framework in any location will inherently be case-specific, and local insights require models calibrated to local conditions. In the Sect. 3 case study significant effort went into finding validation data for the Sonoma County study area, both for the overall component model fit/calibration and for the specific case study events under consideration. The datasets collected as part of this project varied widely both in spatial and temporal resolution and in ease of access. Atmospheric information, precipitation, and streamflow timeseries data were all readily available from academic or governmental institutions. Most of these institutions were involved not only in data collection, but also in the curation and maintenance of web architectures that allowed easy navigation of these datasets. Gridded maps of inundation extents, while not available for many historical events, are becoming more prevalent as remote sensing moves this particular task from one that must be completed on-the-ground to one that can be automated and released in near real time. Damage and loss data, though, are still primarily recorded at the local level through surveys, home inspections, and other resource intensive methods. Difficulties surrounding data collection and individual privacy concerns mean that there are far less publicly available data for analyses of these steps of the model sequence. As a consequence, moving from inundation to damage and from damage to loss are the most uncertain aspects of flood risk assessment due to large uncertainties in the physical mechanisms and the documented difficulties in validating against observed data (Apel et al., 2009; Gerl et al., 2016). Therefore these would be the hardest component models to implement in a new location if the PARRA framework is applied elsewhere.

Two potential pathways could help to close this data gap. First, insurance companies often have extensive information about flood damage and loss at the household level, but due to the legal challenges and the need to retain competitive market advantage there are few opportunities for access. New models for academic-industry data sharing partnerships could help further research in this area. Second, Sonoma County, like a growing number of cities and counties nationwide, maintains an open data portal that proved to be invaluable to this project. Expanding the quantity of these data offerings, and adding damage and loss estimation data at the local level, would improve the outputs of analytical modeling efforts and in turn produce more relevant findings for community level decision making. In return, implementing the PARRA framework with fine-resolution local data provides communities with much more relevant information than they would be able to gain from a large-scale regional or global flood risk assessment.

## 5.3 Component Model Alternatives

The case study developed in Sect. 3 shows only one possible implementation choice for each component model. All models are imperfect representations of the physical world, and there will always be some nuance lost when moving from theory (the framework) to practice (the implementation). There are multiple possible methods to characterize some of the pinch points that would improve fidelity to the underlying physical processes but would increase computational demand and therefore constrain the representation of the true uncertainty. These inherent tradeoffs between different types of error are unavoidable, but point to

a major strength of the PARRA framework: that the user is able to explicitly define their own optimization criteria and choose the component model implementations that best suit their personal expertise, resource constraints, and end goals.

Another core strength is the framework's modularity, meaning that the model chain does not need to be recalibrated or built from scratch to examine a different implementation choice. As long as connections between the pinch points are maintained then any of the component model choices can be modified or replaced with relatively little overhead. For example, the linear regression used for the precipitation component model implementation could be replaced with a numerical weather prediction scheme that explicitly captures the effects of additional factors such as AR orientation. The precipitation pinch point variable itself could be modified to represent the full spatial distribution of precipitation rather than an areal average. The depth-damage relationships used for the damage model could be replaced with an assembly-based vulnerability model, such that the damage to a building is measured as the sum of all damages to its component parts. Many substitutions and modifications to the component model implementations are possible and could be realized without changing the underlying structure or the output metrics produced by the PARRA framework.

## 6   Conclusions

This paper introduced the Performance-based Atmospheric River Risk Analysis (PARRA) framework to quantify AR-induced flood risk. The framework captures the physical processes connecting atmospheric forcings, hydrologic impacts, and economic consequences of AR-driven fluvial flooding. Using a performance-based engineering paradigm, this approach offers several benefits. It quantifies the uncertainty surrounding the physical processes by following a deliberate, ordered simulation procedure. It connects multiple physical processes in sequence by constructing a chain of discrete component models that link together at defined pinch points. Pinch points in the model chain serve to facilitate intercompatibility across different disciplines and to better understand the complexity of the hazard and risk.

Section 3 discussed the fit and calibration of five individual component models: precipitation estimation, hydrologic routing, inundation modeling, depth-damage relationships, and loss estimation. We demonstrated the uncertainty quantification capabilities and the modularity of the component models through case studies of historic AR events affecting the lower Russian River in Sonoma County. We performed step-by-step comparisons between each of these component models and ground-truth data from the case study AR events to show how the differences between the observed and simulated values produced new insights about what drove certain events towards extreme consequences and not others.

In Sect. 4, we ran a fully probabilistic simulation of a damaging February 2019 AR event using only the observed AR characteristics and antecedent soil moisture as input and examined both the probabilistic range and the spatial distribution of the predicted losses. We also used the PARRA framework to generate a first-of-its-kind loss exceedance curve for the lower Russian River to understand the full spectrum of potential loss events rather than a single scenario event or a long-term annual average. We quantified the reduction in flood risk from a hypothetical mitigation decision: when 200 homes were elevated above the 100-year flood elevation the AAL was reduced by half, and the average benefit for events with return periods of 100

years or longer was found to be $50-$75 million per AR. Section 5 highlighted additional nuances about implementation and validation for potential users of the PARRA framework to consider.

While the case study showed examples of the specific insights that can be gained from implementing the component models for a community risk assessment, the theory and scientific merit of the PARRA framework stand on their own, independent from the specific benefits and tradeoffs inherent in any local implementation. We have proposed a new method for the structured assessment of AR-driven flood risk that is physically based, modular, probabilistic, and prospective. The PARRA framework is ideally suited to perform a forward-looking evaluation of potential impacts for events outside of the historic record, or events that have not yet occurred but could in an evolving climate. It can similarly be used to estimate changes in future flood risk due to land use shifts, population change, and more. The framework presented here has been shown to work in a real world implementation and has the potential to greatly expand our understanding of the risks associated with AR-induced flooding.

*Code and data availability.*

Readers are encouraged to explore the supplemental code release associated with this paper, available at https://doi.org/10.5281/zenodo.5765811.

All data used for the historic catalog and for the case study are free and publicly available. Selected sources are included below.

Watershed boundaries and hydrography information were retrieved from the USGS National Map at https://apps.nationalmap.gov/downloader/.

Census boundaries and designations were retrieved from the US Census Bureau (https://www.census.gov/geographies/mapping-files/time-series/geo/tiger-line-file.html) using the R package *tigris* (https://CRAN.R-project.org/package=tigris).

Precipitation data came from the CPC Hourly US Precipitation dataset, provided by the NOAA/OAR/ESRL PSL in Boulder, CO and retrieved using the R package *rnoaa* (https://CRAN.R-project.org/package=rnoaa).

Streamflow data was retrieved from the USGS National Water Information System (http://dx.doi.org/10.5066/F7P55KJN) using the R package *dataRetrieval* (https://code.usgs.gov/water/dataRetrieval).

The bare earth digital elevation model for the study area was retrieved from Sonoma Veg Map at http://sonomavegmap.org/.

The 100-year return period streamflow at USGS gage 11463500 (study area inlet) was calculated using the USGS Stream-Stats application at http://streamstats.usgs.gov/.

The 100-year calculated inundation extent for Sonoma County was retrieved from the FEMA National Flood Hazard Layer at https://www.fema.gov/flood-maps/national-flood-hazard-layer.

Predicted inundation maps for the lower Russian River are available from the County of Sonoma at https://sonomacounty.maps.arcgis.com/home/item.html?id=9d8d63558c6b4124b000e6476a0a020d.

Building footprints are available from the County of Sonoma at https://sonomacounty.maps.arcgis.com/home/item.html?id=0f5982c3582d4de0b811e68d7f0bff8f.

Building occupancy classes and valuations are available from the County of Sonoma, Clerk Recorder Assessor at https: //sonomacounty.maps.arcgis.com/home/item.html?id=2202c1cd6708441f987ca5552f2d9659.

Rapid Evaluation Safety Assessment (RESA) tag information following the 2019 flood event in Sonoma County was retrieved from Permit Sonoma at https://sonomacounty.ca.gov/PRMD/Eng-and-Constr/Building/RESA-2019-Flooding/.

Median values of owner-occupied housing units by census tract were retrieved from the American Community Survey using the R package *censusapi* (https://cran.r-project.org/web/packages/censusapi/vignettes/getting-started.html).

Information about National Flood Insurance Program claims and policies and about the FEMA Hazard Mitigation Assistance program was retrieved from the OpenFEMA data platform at https://www.fema.gov/about/openfema/data-sets.

All figures with color were generated using the *roma* scientific color map from Crameri et al. (2020).

*Author contributions.* CCB: Conceptualization, Data curation, Formal analysis, Methodology, Validation, Visualization, Writing - original draft, Writing - review & editing. KAS: Conceptualization, Methodology, Supervision, Writing - review & editing. JWB: Conceptualization, Methodology, Resources, Supervision, Writing - review & editing.

*Competing interests.* The authors declare that they have no conflict of interest.

*Acknowledgements.* This material is based upon work supported by both the Stanford Gabilan Graduate Fellowship and the National Science Foundation Graduate Research Fellowship under Grant No. 1000265549. Any opinion, findings, and conclusions or recommendations expressed in this material are those of the authors and do not necessarily reflect the views of the National Science Foundation. Some of the computing for this project was performed on the Sherlock cluster. We thank Stanford University and the Stanford Research Computing Center for providing computational resources and support that contributed to these research results. We also thank the two anonymous reviewers for their constructive feedback which has greatly improved the quality of this paper.

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
