# Peer review of "A Performance-Based Approach to Quantify Atmospheric River Flood Risk"

_Natural Hazards and Earth System Sciences, 2021_

## Referee Comment (RC2)

This paper provides a framework (PARRA) to quantify Atmospheric River flood risk using performance-based submodules for Sonoma County, CA. The methodology is described in detail, and a case study from a 2019 AR event was investigated. The PARRA framework is very interesting and useful for providing a mean estimate of expected losses with uncertainty bounds. However, I have some concerns about the methodology and the way it was described in the manuscript. I also have some concern about the 2019 case study that was investigated – why choose a case study that the PARRA framework barely captures in the tail of its distribution? Why not show a case study that the PARRA framework captures much better? I have some comments below that can help improve the paper. I think the paper can be accepted after some minor revisions and clarifications.

General Comments

What would the PARRA framework provide a stakeholder as the estimated losses for the 2019 AR event? The results show an average loss of about $25 million (Figure 9a), but the actual cost was $91.6 million. This actual cost is covered in the tail of the distribution provided by PARRA but is far from the mean of this distribution. Do the authors consider this an accurate assessment? Some comments on how to interpret the results with their associated uncertainties, as well as how to interpret the acceptability of the results, would be helpful.

Specific Comments / Technical Corrections

Section 1 Introduction

Line 27: Pineapple express is not the only mechanism that brings ARs to California.

Line 56: "… understanding climatology of ARs", see Espinoza et al. 2018 and Massoud et al. 2019 who aimed to understand AR climatology in a global context.
  (a) Espinoza, Vicky, Duane E. Waliser, Bin Guan, David A. Lavers, and F. Martin Ralph. "Global analysis of climate change projection effects on atmospheric rivers." Geophysical Research Letters 45, no. 9 (2018): 4299-4308.
  (b) Massoud, E. C., V. Espinoza, B. Guan, and D. E. Waliser. "Global climate model ensemble approaches for future projections of atmospheric rivers." Earth's Future 7, no. 10 (2019): 1136-1151.

Section 2 Framework Description

Line 110: Is this theorem a version of Bayes theorem? How are they related?

Line 143: Initially it seems that the AR category score (1-5) is used as input in the PARRA framework. It isn't until later in the manuscript that it becomes clear that AR max IVT and duration are used. The authors should clarify this earlier in the paper.

Line 147: Some precipitation can be from non-AR sources. Is this considered for the calculation of the precipitation submodule in the PARRA framework?

Line 160: take out the word 'are'

Section 3 Case Study: Sonoma County

Line 235: Is there a citation that shows why WLS can be used to express the relationship between IVT/DUR and PRCP? This seems rather simplistic and not thorough enough to capture the estimated PRCP. According to Figure 4 there seems to be significant spread in these relationships. Perhaps the authors can explain why this choice was made.

Line 260: The mean of the distribution is way off here. Should this be a reason for concern? It seems that this methodology begins to break down for extreme AR events.

Line 266: "… a suitable representation of reality", this is a subjective acceptance criterion, and the authors should note it as so.

Line 282: Figure 6 is mentioned before Figure 5.

Line 332/Figure 5: The initial peak on Feb 26 is not captured. Can the authors provide some comments and reasoning behind this?

Line 339: This comment is applicable for this section and for other sections. There are several choices that need to be made by the user, such as the LISFLOOD parameters. This raises the question of the PARRA method's applicability to other locations. Does the whole framework need to be re-calibrated with local data for other local case studies?

Line 352: There is no information on which surrogate model was used, and what the accuracy or efficiency of that surrogate model is. In general, there is very little information on this emulation method or how it is used. How can other readers re-produce or build on this analysis if this critical information is missing?

Line 440: On correction factors - Again, this seems like a subjective fix for applying the PARRA framework in this region. How can the framework be applied elsewhere using this methodology? Although the framework seems to be useful for Sonoma County, how can the authors show that the methodology can still be efficiently applied for other locations? Some ideas that address this question can be helpful in accepting the PARRA framework as a generally usable framework.

Figure 9: The distribution just barely captures the observed event in its tail. As mentioned above, how is this result with its uncertainty reported to a manager or a stakeholder? What is the provided answer here?

Section 4 Results

Line 477: Equation 6 - Is this equation reported anywhere in the literature? Seems like another subjective criteria that the authors implement. There needs to be more information describing this choice.

Line 482: The AAL is an interesting concept to describe the average annual losses. However, it is known that this region experiences significant swings between wet and dry years. Is it feasible for the authors to calculate what the AAL is for wet vs dry years?

Line 487: What are the uncertainties around these estimates? Do the authors provide this?

Line 522: the word 'the' is duplicated

Line 524: 'Expected benefits' - See Massoud et al. 2018, who did a similar analysis for groundwater and investigated how changes to decisions in managing water resources can impact expected changes to groundwater storage. Studies like this are starting to populate the literature.
  (a) Massoud, Elias C., Adam J. Purdy, Michelle E. Miro, and James S. Famiglietti. "Projecting groundwater storage changes in California's Central Valley." Scientific reports 8, no. 1 (2018): 1-9.

Line 529: Take out the words 'is of the 2019'.

Section 5 Discussion

Line 535: I would argue that these insights are helpful for planners, managers, and engineers, yet not so helpful for purely scientific investigation since many choices in the framework are purely subjective. I think it is important for the authors to make this clear throughout the paper.

Line 543: Another process that can matter here is the role of sequential ARs (i.e., multiple ARs occurring sequentially), something to consider for 'future directions'.

Line 557: Yes, but what did this do to the expected accuracy of capturing the relationships? The framework is trading potential accuracy and confidence for computational efficiency. This introduces even more uncertainty. The authors should state this.

Line 582: Component Model Alternatives - This is where some of the subjective choices of the framework can be replaced with more objective choices, and therefore can make the framework more sound for scientific analysis.

Line 589: the word 'the' is duplicated

Line 591: the word 'underlying' is duplicated

Section 6 Conclusions

Line 594: Is it possible/feasible to test another case study event that the PARRA framework accurately estimates the damages for? This can help show case the value of the PARRA framework even more than just showing the one case study from 2019 that was barely captured in the tail of the distribution.

Line 608: '… event fell within the expected probabilistic range …', In the tail of the distribution. It was barely captured. The authors should be careful with how they communicate the accuracy of the provided result.

---

## Author Comment (AC1)

**Authors response to Anonymous Referee #1**

Referee comment on "A Performance-Based Approach to Quantify Atmospheric River Flood Risk" by Corinne Bowers et al., Nat. Hazards Earth Syst. Sci. Discuss., https://doi.org/10.5194/nhess-2021-337-RC1, 2021

We would like to thank the reviewer for their valuable feedback and constructive comments. We have revised the manuscript accordingly and provided detailed responses to each of the comments below.

General Comments

**1.1)**
The authors provide a process-based probabilistic framework for predicting damages associated with ARs based on AR intensity and duration and antecedent hydrologic conditions. This is a useful tool. There are a number of technical innovations throughout the study. The 2019 Russian River case study contains several creative data sourcing and manipulation steps to overcome inherent data availability issues. The overall method has broader applications than AR damage prediction. It could be applied to any damaging hydrometeorogic events, including hurricanes and tropical storms. The AAL and loss exceedance curve calculations are compelling. A number of valuable insights are presented in the discussion section.

Response: We would like to thank the reviewer for these comments about our work and its potential applications.

**1.2)**
My only comment of substance is that the current ordering of sections makes the model difficult to follow as variables are introduced before being defined and the multivariate Monte Carlo integration framework is explained after presenting the series of integrals. I'd put section 2 paragraph 1 first, then 3.2 paragraph 1, then 2.1 description of pinch point variables, then 2 framework description with the equations, then 3.2 paragraph 2 explanation of Monte Carlo integration. Or something along those lines. The current ordering was difficult for me to follow although it did all make sense at the end.

Response: We acknowledge that the original ordering created unnecessary confusion and have reorganized much of Section 2 to address the reviewer's concerns. We appreciate the helpful suggestions on how to present the information in a more intuitive order. The new organization is as follows:

Section 2: Framework Description

- Law of total probability (same)
- Introduction of Eq. 1 (modified to address comments 1.8 and 1.15)
- Definition of pinch points (previously the first paragraph of Sect. 2.1)
- Definition of component models (previously the first paragraph of Sect. 2.2)
- Introduction of Eq. 2 (modified to address comment 1.15)
- Explanation of Monte Carlo integration (previously the second paragraph of Sect. 2.2)

Section 2.1: Pinch Point Variables
- Distinction between pinch points and pinch point variables (new; added to address comment 1.14)
- Detailed description of pinch point variables (same)

Section 2.2: Component Models
- General description of component models, with special attention to $f(Q \mid PRCP, HC)$ (new; added to address comments 1.13 and 1.36)
- Distinction between component models and component model implementations (new; added to address comments from Reviewer 2)

We believe that this response serves to additionally resolve the concerns and questions expressed by the reviewer in comments 1.12, 1.16-1.24, 1.31, and 1.32.

**1.3)**
The rest of the comments are minor technical corrections / suggestions or requests for clarification. Overall this is a great contribution to the literature. I recommend accepting the manuscript after minor revisions.

Response: We would like to thank the reviewer for their recommendation, and for the time and effort put into reviewing this manuscript.

- - - - -

Specific Comments / Technical Corrections

1 Introduction

**1.4)**
Line 26 California experiences ARs coming from a pathway called the Pineapple Express -> California often experiences ARs coming from a pathway called the Pineapple Express [not all ARs in CA are considered Pineapple Express storms]

Response: We acknowledge the reviewer's suggestion and we have removed this sentence from the manuscript.

**1.5)**
Line 31 $300 million -> $660 million
Data appendix S1 Top Counties lists damages for CA counties over 40 years at 26.53 billion of which AR damages were 24.86 billion in 2019 dollars. This translates into annual AR damages of 24.86/40 = 621.5 million. In 2021 dollars this is approximately $660 in 2021 dollars (e.g., https://www.minneapolisfed.org/about-us/monetary-policy/inflation-calculator)

Response: We have revised according to the reviewer's suggestion.

1.1 Disciplinary Context

**1.6)**
In addition to FEMA's Hazus, USACE's HEC-FIA and HEC-FDA are potential methods that can be used to convert HEC-RAS outputs to economic impacts.

Response: We acknowledge the reviewer's suggestion and have added the following sentence to Line 72.

*"Other regional flood loss assessment tools include HEC-FIA (USACE, 2018) and HEC-FDA (USACE, 2014), both from the US Army Corps of Engineers; and FloodFactor (Bates et al., 2020), a commercial product from First Street Foundation."*

2 Framework Description

**1.7)**
Line 109 total probability theorem -> law of total probability

Response: We have revised according to the reviewer's suggestion.

**1.8)**
Line 116 decision variable DV appears here for the first time but is described later at 165. Either move the section on pinch point variables above the introduction of DV, etc., or note that the variables are defined in detail below.

Response: We acknowledge the reviewer's comment, and we have added the following text to Line 114 to indicate that an explanation of the pinch point variable DV will be provided later in the manuscript.

*"We first replace the generic variables with new variables representing pinch points, which we elaborate on later in this section. B becomes the atmospheric river event AR and A becomes the decision variable DV."*

**1.9)**
Eq 2 consider \cdot or \times in place of asterisks.

Response: We agree with the reviewer and have switched to using \cdot throughout the manuscript.

**1.10)**
Eq 2 consider six integrals evenly spaced rather than two sets of three integrals.

Response: We have revised according to the reviewer's suggestion.

**1.11)**
Eq 2 it would perhaps make sense to include the supports over which the pinch point variables are being integrated. But perhaps it would be a distraction.

Response: We tested out the reviewer's suggestion, but felt that adding the integration supports created visual clutter and reduced the clarity of the equation. We have thus retained the integrals without explicit supports.

**1.12)**
Eq 2 ideally, the variables should be defined as they are introduced. Perhaps it is sufficient to note that the variables are defined below in Sect. 2.1.

Response: We have reorganized Sect. 2 as outlined in our response to comment 1.2 to address this suggestion.

**1.13)**
Eq 2 An explanation of why $f(Q \mid PRCP, HC)$ has two conditional variables while all other elements in the chain have only one may be useful for the reader, at some point in the text.

Maybe something like: at each point in the causal chain one pinch point variable depends on the next. Flow, Q, depends on two variables precipitation, PRCP, and antecedent hydrologic conditions, HC. Could perhaps write out the whole chain in English in the paragraph below the equation. This would be easier to follow than waiting to read the text in the next section. Or else note that all variables are defined in the following section.

Response: We acknowledge the reviewer's comment, and we thank the reviewer for providing some suggested text to improve the manuscript. We have reorganized Sect. 2 as outlined in our response to comment 1.2 to provide the reader with the component model definitions before the equation. We have also added an explanation of why f(Q | PRCP, HC) has two conditional variables to the new version of Sect. 2.2. The text of this new paragraph is as follows.

*"The pinch points in the framework are linked by component models, or representations of discrete physical processes. Each component model generates an expected distribution of values for the next pinch point in the sequence conditioned on the value(s) preceding it. It is important to note that, excepting the hydrologic routing model f(Q|PRCP,HC), all models are conditioned on only one variable. The hydrologic routing model differs from the others because, like the event characteristics (AR), the antecedent hydrologic conditions (HC) are framework inputs provided by the user to represent an initial system state. All other pinch point variables represent calculated variables. Conditioning on a minimal number of variables is critical to achieving the objective of modularity because it reduces the data demands at each step of the modeling process."*

**1.14)**
Eq 2 Each pinch point variable is a scalar here?

Response: We have reorganized Sect. 2 as outlined in our response to comment 1.2 to provide the reader with the pinch point definitions before the equation. We have also added additional text to distinguish between pinch points and pinch point variables, as outlined below. The revised Sect. 2.1 includes the following paragraph (approximately Line 140 in the manuscript).

*"The pinch points presented in Sect. 2 are conceptual descriptions of the intermediate system states between AR occurrence and flood loss where only a limited amount of information must be transferred to the next step. Pinch point variables are low-dimensional numerical vectors of the information at each pinch point (Garrick 1984). The following paragraphs expand upon the conceptual pinch points and introduce the specific dimensions and measurement units that are used in this paper for each pinch point variable."*

**1.15)**
Eq 2 I'm unclear on how \lambda(AR) works and on how \lambda(DV > x) and P(DV > x | DM) work... An additional line providing some context could be helpful.

Response: We thank the reviewer for drawing attention to these elements of the PARRA framework, and we have modified the descriptions of both Eq. 1 and Eq. 2 as a result. For Eq. 1, we have removed the text from Lines 113-121 in the manuscript and replaced it with the following.

*"Equation 1 modifies the statement of the law of total probability to better fit the context of natural hazard assessment.*

$$\lambda(DV > x) = \int P(DV > x \mid AR) \cdot \lambda(AR) \, dAR \quad (1)$$

*where $\lambda(DV > x)$ is the rate of the decision variable DV exceeding some specified threshold x, i.e., how frequently losses exceed \$x dollars; $P(DV > x \mid AR)$ is the probability of DV exceeding x conditioned on an inducing AR event; and $\lambda(AR)$ is the occurrence rate of that inducing event. The right side of the expression is integrated over all possible inducing events in the sample space. We evaluate $\lambda(DV > x)$ at a range of x values to obtain the loss exceedance curve, which is explored further in Sect. 4.2.*

*"We first replace the generic variables with new variables representing pinch points, which we elaborate on later in this section. B becomes the atmospheric river event AR and A becomes the decision variable DV. $P(DV > x)$ is the complement of the cumulative distribution function for DV, starting at 100% probability of exceedance for low values of x and moving to a probability of zero as x increases. $P(DV > x \mid AR)$ represents the probability of the decision variable DV exceeding some threshold value x conditioned on the inducing event AR.*

*"We then transform the summation into an integral and move to calculating the occurrence rate $\lambda$, which represents a continuous state variable rather than the probability P of a discrete event. Probabilities are defined with respect to predetermined time periods, and the probability of seeing an AR event in the next week, month, or year are all different quantities. Calculating the occurrence rate $\lambda$ offers similar information about the underlying phenomenon of interest (AR event frequency) without imposing an arbitrary time limitation."*

For Eq. 2, we have removed the text from Lines 126-131 in the manuscript and replaced it with the following.

*"...where variables AR, PRCP, HC, Q, INUN, DM, and DV represent pinch points and the conditional probability expressions represent component models. The component models of the form $f(Y \mid X)$ are conditional probability density functions that describe the distribution of results from numerical analyses. The component model $P(DV > x \mid DM)$ measures the probability of pinch point DV exceeding the loss threshold x conditioned on DM. The PARRA framework is executed by starting with the outermost integration in the equation and moving inward, as each component model is conditioned on the one(s) preceding it in the model chain. This equation is also represented visually in Fig. 1."*

**2.1 Pinch Point Variables**

**1.16)**

(The following comments were written as I was reading through the manuscript. They could be avoided if you are clear upfront about how the integration is Monte Carlo integration and how the pinch point variables can be vectors.)

Response: We agree with the reviewer's overall comments on the organization of Sect. 2, and we have reorganized the section as outlined in our response to comment 1.2. We believe that comments 1.17 through 1.23 have been resolved by this reorganization and by the additional understanding the reviewer has noted from comment 1.24 onward.

**1.17)**

I'm a little unclear on how the causal chain works with scalar variables given that the process is spatially heterogeneous. Should I think of the PARRA process running in parallel over all locations? But what about spatial correlations?

Response: Please see our response to comment 1.16.

**1.18)**

AR is a measure of intensity, so it could be something like peak IVT or cumulative vapor transport over some time period, the duration of the AR, say. But IVT is a vector field. So you'd need to aggregate or average over time and space to get a scalar metric of intensity.

Response: Please see our response to comment 1.16.

**1.19)**

PRCP as a scalar field has the same issue. You'd need to integrate over time and space to get a scalar value. Should I think of this as some metric of precipitation over the whole watershed? Or is there a way to apply PARRA with a time series of precipitation grids as inputs?

Response: Please see our response to comment 1.16.

**1.20)**

HC, same as AR and PRCP.

Response: Please see our response to comment 1.16.

**1.21)**

Q makes more sense as a single input if you're considering a single channel, although the hydrograph is a curve which captures the duration as well as the intensity of the flow above flood stage, so I'm unclear on how this enters into the formulation.

Response: Please see our response to comment 1.16.

**1.22)**
INUN at a given location or structure is just a scalar, but over a set of n structures is an n-dimensional vector. Here, duration of inundation may also be important, in addition to depth, in terms of generating damages.

Response: Please see our response to comment 1.16.

**1.23)**
I'm unclear on how DM and DV differ. DV is a metric of impact or consequence. DM is a damage measure. So, DV could be a more broad measure of impact that is perhaps related to DM through some probabilistic relationship that is modeled using the observational record?

Response: Please see our response to comment 1.16.

**1.24)**
Ah, the variables are discussed in more detail. AR is a vector of max IVT and duration, got it.

Response: We thank the reviewer for identifying where this portion of the manuscript achieves clarity of purpose.

**1.25)**
PRCP is storm-total accumulated rainfall over the watershed. Did you experiment at all with more complex formulations for precipitation? Don't tools like HEC-RAS and LisFlood take precipitation fields as inputs?

Response: We decided to present a simpler scalar representation of precipitation to streamline the model chain and keep the focus of the manuscript on the overall framework rather than the individual component model implementations. While we agree with the reviewer that both HEC-RAS and LISFLOOD have the computational capability to accept two-dimensional precipitation fields, the complexity of the precipitation pinch point variable was limited not by these dynamical models but by the statistical model chosen to represent the component model f(PRCP|AR) described in Sect. 3.2.1. We discussed some of these tradeoffs later in the manuscript, starting at Line 585, and we have added the following text to Line 150 to point readers to that discussion.

*"This simplification could be modified or eliminated in future implementations of the precipitation component model, as addressed further in Sect. 5.4."*

**1.26)**
HC watershed-average soil moisture equivalent height. There's probably some additional uncertainty introduced by averaging over the whole watershed. Upstream soil moisture may be more relevant than downstream soil moisture, for example, although these are probably highly correlated.

Response: We have modified the sentence at Line 123 to reflect that soil moisture values were averaged across only the upstream watershed, not the entire watershed. The revised sentence is as follows: *"Therefore the pinch point variable representing antecedent hydrologic conditions is a scalar value measuring the average soil moisture in the upstream watershed."*

We also acknowledge the reviewer's comment about the introduction of additional uncertainty through averaging. Because of the relatively low temporal and spatial resolution of the CPC soil moisture dataset (1 month and 0.5 degrees lat/long, respectively), we found that additional precision in averaging did not significantly affect the values calculated for the events in the historic catalog.

**1.27)**
Q is time series of flow at inlet. This is parameterized as a 3-vector with Q_p, t_p, and m.

Response: We appreciate the reviewer's comment.

**1.28)**
INUN is surface water depth at locations of interest. So this is N-dimensional.

Response: We appreciate the reviewer's comment.

**1.29)**
DM is a damage ratio, expected cost to repair over the total value. Assumed to be a function of water depth.

Response: We appreciate the reviewer's comment.

**1.30)**

DV actionable measure of impacts. So, it converts damage ratios into damages? So it requires observed building values then? What's the utility in splitting DM and DV? I think I can see it, but an explanation could be useful.

Response: We thank the reviewer for identifying the need for more information about the distinction between these two pinch point variables. We have revised the last paragraph of Sect. 2.1 and added the following additional text to Lines 166-168.

*"Finally, the decision variable (DV) is some actionable measure of AR impacts. In this work we define DV as household-level monetary losses; however, DV could alternatively represent any other metric that is calculated as a function of the damage measure DM, such as the number of displaced persons or the time to full recovery."*

2.2 Component Models

**1.31)**
This could perhaps go above the equations.

Response: We have reorganized Sect. 2 as outlined in our response to comment 1.2 to address this suggestion.

**1.32)**
I'd put section 2 paragraph 1 first, then 3.2 paragraph 1, then 2.1 description of pinch point variables, then 2 framework description with the equations, then 3.2 paragraph 2 explanation of Monte Carlo integration. Or something along those lines. The current ordering was difficult for me to follow.

Response: We have reorganized Sect. 2 as outlined in our response to comment 1.2 to address this suggestion.

3 Case Study: Sonoma County

**1.33)**
Line 185 The spatially repetitive, locally severe flooding seen in Sonoma County is a signature characteristic of ARs. <- I'm not sure if I agree with this statement; I suggest removing it. The statement suggests that ARs tend to reoccur at the same locations and always generate locally severe flooding. Some ARs generate multi-basin flooding, like the 1862 event. Some locations affected by ARs flood (relatively) infrequently.

Response: We agree with the reviewer and have removed this sentence from the manuscript.

3.2.1 Precipitation Component Model

**1.34)**
Line 238 mixture model 90% with WLS standard errors, 10% with distribution fit to largest 10% of events. I'm familiar with WLS but not with this approach. More detail on this method, or a reference, would be helpful.

Response: We have adopted a Gaussian mixture model to represent the non-normal residuals in both the precipitation and streamflow regressions. We used mixture models where the regression coefficients are held constant and only the errors are allowed to vary, and we further restricted the errors to be represented by a mixture of zero-mean Gaussian distributions such that only the differences were the error variances. We chose the Gaussian mixture model because it is semi-parametric, meaning it is more flexible than other parametric distributions that could be used for non-normal errors, and because mixture models are often applied when we believe that there are latent variables or that observations may be coming from different populations (i.e., precipitation stemming from different climatological drivers). We have added citations to Line 240 referencing related work by Bartolucci and Scaccia (2005) and Soffritti and Galimberti (2011) for more information on this method.

3.2.2 Precipitation 2019 Case Study

**1.35)**
Line 271 We note that Sonoma County is not guaranteed to see any impacts -> We note that, according to the simulated distribution, Sonoma County... (or, according to the distribution simulated from the observational record, etc.)

Response: We have revised Line 271 according to the reviewer's suggestion. The new sentence reads: *"Our simulation results indicate that Sonoma County is not guaranteed to see any impacts…"*

3.3.2 Hydrologic Conditions 2019 Case Study

**1.36)**
Line 289 it is interesting that soil moisture is an "input" here and not simulated, just as AR IVT and duration are "inputs." This is explicitly captured in the flow chart and in the Eq 2 multiple integral. It might be worth emphasizing this in the description of the flow chart, for example.

Response: We appreciate the reviewer's suggestion. We have highlighted the fact that soil moisture is an input rather than a calculated variable in the new version of Sect. 2.2, as outlined in our response to comment 1.2. The text of this new paragraph is included as part of our response to comment 1.13. We have also modified the caption of Fig. 1 as follows:

*"Figure 1: PARRA framework flowchart. Graphical depiction of the PARRA framework, as presented mathematically in Eq. 2. White boxes represent component models. Arrows represent pinch points: an arrow pointing towards a box indicates a required component model input, and an arrow coming out of a box indicates a component model output. The background colors broadly represent existing research domains."*

We would additionally like to note that this comment from the reviewer brought a numerical error to our attention in the calculation of soil moisture, which has affected both the estimate of losses for the 2019 event and the overall AAL for the study area. We have included the updated results as part of our response to comment 1.49.

**1.37)**
Line 291 why is observed precipitation used as an input here? Shouldn't the full precipitation distribution, derived from the input AR intensity and duration, enter here? What am I missing?

Response: We thank the reviewer for noting that this sentence created confusion. The observed precipitation and soil moisture are used as input for the streamflow component model in Sect. 3.4 because this section focuses on model-by-model calibration and comparison. To resolve confusion for future readers, we removed the sentence referenced by the reviewer and added a new sentence to Line 326 to improve clarity. The introduction to Sect. 3.4.2 now reads as follows:

*"Given the 2019 observed precipitation and antecedent soil moisture, we generated 1,000 Monte Carlo realizations from the streamflow model and compared the predicted streamflow hydrograph from the calibrated component model implementation to the observed hydrograph from the February 2019 event. Using observed data as input rather than the simulated distributions from Sects. 3.2 and 3.3 allows us to examine the fit and uncertainty associated with this specific step of the model chain in isolation."*

3.4.1 Flow Component Model

**1.38)**
Line 310 - what data were you using here? The observational precipitation record? Fed into the runoff calculation? So, you have how many observations to fit the mixture OLS model?

Response: The OLS regression referenced by the reviewer predicts the peak streamflow value $Q_p$ as a function of precipitation and runoff. The coefficients of the regression were fit based on observed precipitation and runoff values from the historic catalog of 382 events. Based on the reviewer's questions presented here we have chosen to shorten the discussion of the runoff calculation, because we felt it was drawing attention away from the main point of the section. The full calculation process is still available through the supplemental code release. We have replaced the text from Lines 302-309 with the following sentence:

*"Runoff, the portion of precipitation that flows over the ground surface rather than contributing to evapotranspiration or infiltration, was calculated for each event in the historic catalog using the empirical curve number method (NRCS, 2004, Chapter 10)."*

3.4.2 Flow 2019 Case Study

**1.39)**
Fig 5 b - any speculation on the early streamflow peak in the 2019 event? It doesn't seem to be captured within the 90% PI. A horizontal line indicating flood stage could also be informative in this figure.

Response: We acknowledge that the early streamflow peak seen in Fig. 5(b) requires more contextualization. We have added a hyetograph of observed precipitation to the figure as shown below. We have also added additional commentary on both the early peak and the new hyetograph. The new text starts at Line 332 and reads as follows.

*"The complex shape of the observed streamflow timeseries in Fig. 5(b) is a function of the unique watershed response as well as the spatial and temporal heterogeneity of the input precipitation. By contrast, the simulated distribution is based on the unit hydrograph method, which assumes that the precipitation distribution is uniform and that all runoff enters the channel at a single location. This limits our ability to capture certain kinds of behavior, such as the early peak seen in the observed streamflow timeseries in Fig. 5(b). The early peak could be due to catchment processes that cause a lagged tributary response, input from direct surface runoff, spatial variation in precipitation intensity and duration, or any number of other mechanisms. We include the observed hyetograph to the top of the plot in Fig. 5(b) to show just one aspect of the natural variability that affects the observed timeseries.*

*"Despite the simplification imposed by the unit hydrograph method, many metrics of interest are reasonably well characterized by the simulated timeseries. The observed peak streamflow (1,130 $m^3s^{-1}$) is at the 43$^{rd}$ percentile and the observed floodwave duration (81 h) is at the 63$^{rd}$ percentile of the respective simulated distributions. Recall from Sect. 3.2 that the observed precipitation was notably high conditioned on the observed atmospheric conditions. We now*

*note that while the observed streamflow may have been high for a Category 3 event, it was in the middle of the simulated distribution conditioned on the observed precipitation. Therefore we conclude that the hydrologic routing was likely not one of the physical processes contributing to the "extremeness" of the 2019 event."*

Additionally, we appreciated the reviewer's suggestion to add a line indicating the flood stage. We have added one to the revised version of Fig. 5 below.

[Figure]

*Revised streamflow figure.*

3.5.1 Inundation Component Model

**1.40)**
Line 344 100 year peak flow -> 100-year peak flow, etc. (make this change throughout the manuscript)

Response: We have made the change from "100 year" to "100-year" throughout the manuscript according to the reviewer's suggestion.

**1.41)**
Line 367 how many buildings were there in your domain? What year were the building footprints taken from?

Response: We have added the following sentence to Line 385 to answer the reviewer's question: *"We used building footprints from 2019 SonomaVegMap LIDAR data and building parcel information from the 2021 Sonoma County Clerk Recorder Assessor to identify 41,000 homes within the study area."*

Citations for both of these datasets can be found in the "Code and data availability statement" at the end of the manuscript, which starts at Line 641.

3.5.2 Inundation 2019 Case Study

**1.42)**
Figure 7 in the Data Type legend it appears that Observed is dashed and Simulated in solid. Making this more clear would be helpful.

Response: We have revised Figure 7 according to the reviewer's suggestion.

3.6.2 Damage Measure 2019 Case Study

**1.43)**
RESA tagging is a fascinating approach.

Response: We would like to thank the reviewer for their comment.

3.7.1 Decision Variable Component Model

**1.44)**
Interesting approach to estimating property values from tax assessments adjusted using ACS correction factors.

Response: We would like to thank the reviewer for their comment.

3.7.2 Decision Variable 2019 Case Study

**1.45)**
Line 451 missing comma after i.e.

Response: We have revised according to the reviewer's suggestion.

**1.46)**
Figure 9 b - it would be useful to have a high-resolution version of this figure in the appendix, or in a data appendix.

Response: Figure 9b is available in high resolution as part of the supplemental code release mentioned on Line 622; specifically, it is part of the markdown file named lossexceedance.Rmd that reproduces Figures 9 and 10. Based on the reviewer's comment, we will additionally add a spreadsheet with the values shown in Figure 9b to our next Github code release.

4 Results

**1.47)**
Eq 6 - consider \cdot or \times in place of asterisk, or no multiplication symbol at all. Same comment throughout equations.

Response: We agree with the reviewer and have switched to using \cdot throughout the manuscript.

4.1 AAL

**1.48)**
Line 487 You could note that $156m is likely to be an overestimate given that the county-wide penetration rates are lower than the penetration rates for properties at risk.

Response: We agree with the reviewer that the $156M reported in the manuscript at a county level is likely an overestimate. Upon revisiting the available data, we found we were able to revise the calculation of the NFIP AAL and estimate both the insurance penetration rate at the census tract level rather than the county level. The revised NFIP AAL is $121M. We have revised the description on Lines 483-487 and modified every instance of the word "county" to "census tract" within this paragraph to reflect the change in the calculation process. This is a far more targeted geographical area and is therefore likely to represent insurance penetration rates (and consequently flood risk) relatively well, and we thank the reviewer for highlighting this opportunity to improve our estimate.

**1.49)**
Line 487 What is the uncertainty around the $163m estimate?

Response: Due to the numerical error mentioned in our response to comment 1.36, the soil moisture component model was incorrectly oversampling from the high (wet) end of the soil moisture distribution. In addition, although the manuscript stated that soil moisture was an input based on observed data rather than a simulated value, the PARRA simulation results reported in Figs. 9 and 10 were based on calculations that were using simulated soil moisture values. Correcting this error increased the expected losses for the 2019 event (because the "observed" soil moisture for this event was very high relative to others in the record) and lowered the overall AAL (because the simulated realizations were rebalanced to include more events with dry antecedent conditions). We have edited the manuscript accordingly, and we thank the reviewer for their comment that led to us finding this inconsistency.

As a result of these changes the mean expected AAL in the study area has been revised to $111M, and through Monte Carlo simulation we have estimated a 90% confidence interval to span from $93M to $133M. Based on the reviewer's question we have added the following text to Line 482.

*"The mean AAL estimated from the stochastic record for AR-induced flood losses to residential structures is $111 million, with 90% confidence that it lies between $93 and $133 million."*

5 Discussion

**1.50)**
There are many valuable insights in the discussion section.

Response: We would like to thank the reviewer for their comment.

---

## Author Comment (AC2)

**Authors response to Anonymous Referee #2**

Review for Bowers et al.

**2.1)**
This paper provides a framework (PARRA) to quantify Atmospheric River flood risk using performance-based submodules for Sonoma County, CA. The methodology is described in detail, and a case study from a 2019 AR event was investigated. The PARRA framework is very interesting and useful for providing a mean estimate of expected losses with uncertainty bounds.

We would like to thank the reviewer for their valuable feedback and constructive comments. We have revised the manuscript accordingly and provided detailed responses to each of the comments below.

**2.2)**
However, I have some concerns about the methodology and the way it was described in the manuscript. I also have some concern about the 2019 case study that was investigated – why choose a case study that the PARRA framework barely captures in the tail of its distribution? Why not show a case study that the PARRA framework captures much better?

Response: We acknowledge the reviewer's concerns about both the results of the 2019 case study and the way it was described in the manuscript, and we thank the reviewer for helping us to significantly clarify our presentation of both. We will address these two concerns separately.

First, we recognize that we did not provide enough support to demonstrate that the extremeness of the 2019 event was well represented by the various component model implementations. We agree with the reviewer that presenting additional case studies helps to strengthen this point for future readers. We have therefore added additional case study events to the discussion of precipitation in Sect. 3.2.2 and to the discussion of losses in Sect. 3.7.2. We have included the new versions of each of these sections below. For readers who want to dig deeper into our results and methodology, we will add results for many more AR events to our next Github code release, as referenced in our response to comment 2.18.

NEW SECT. 3.2.2

*"We present a comparison of observed vs. simulated precipitation values for four AR events. Figs. 5(a-b) are the two most recent Category 3 (strong) ARs in the historic catalog, and Figs. 5(c-d) are the two most recent Category 5 (exceptional) ARs. The dashed lines mark the recorded precipitation totals for each event and the tick marks along the top of the panel show the recorded totals from all ARs in the historic catalog in the same intensity category. For each*

event we generated 1,000 Monte Carlo realizations of precipitation given the observed maximum IVT and duration and plotted the resulting distribution as a histogram. The histograms represent realizations of potential precipitation if another AR occurred in Sonoma County with the same characteristics. We do not expect the observed dashed lines to fall in the center of the simulated distributions; rather, the observed values can be thought of as random samples from the simulated distributions, and comparing the two offers new insights into the character of specific ARs.

"For example, Figs. 5(a) and 5(c) show two impactful storms for Sonoma County, from February 2019 and January 2017, respectively. The February 2019 event caused approximately $155 million in total damage (Chavez, 2019) and the January 2017 event caused approximately $15 million (County of Sonoma, 2017). While both events had precipitation totals in excess of 200 mm, the precipitation relative to the event-specific maximum IVT and duration was far higher in February 2019 than January 2017. The January 2017 event was a Category 5 AR, meaning it had the greatest potential for hazardous impacts. Conditioned on the intense atmospheric conditions, though, the observed precipitation was near the mean of the simulated distribution in Fig. 5(c).

"Figure 5(a) shows the predicted precipitation distribution for a Category 3 AR (mixture of beneficial and hazardous impacts) with the same maximum IVT and duration as was observed in February 2019. By all accounts, though, the February 2019 event was a very hazardous storm with severe impacts for communities in the study area. The observed precipitation is at the upper tail of what we would expect for a Category 3 event in both the observed distribution (shown in the tick marks at the top of the plot) and the simulated distribution (shown in the histogram). Therefore we infer that the precipitation is likely one of the drivers that led this particular AR to become a damaging event. In summary, the PARRA simulation results provide evidence that the January 2017 event was a moderate precipitation conditioned on extreme AR hazard while the February 2019 event was an extreme precipitation conditioned on more moderate AR hazard. These are two distinct pathways ARs can take to generate significant consequences.

[revised manuscript text omitted]

*information is known about the storm other than the maximum IVT, duration, and soil moisture. However, AR forecasts now typically include an estimate of the expected regional precipitation total. If we had perfect information about total precipitation (i.e., we could predict in advance exactly what the observed value would be) we could start the PARRA framework at the precipitation pinch point variable PRCP and run all subsequent component models in the sequence probabilistically. Figure 9(c) is therefore an exploration of a "what-if" scenario where losses are conditional on the observed precipitation value from February 2019 rather than the AR characteristics. While the observed $91.6 million loss estimate is similarly extreme in this case (91st percentile event vs. 89th) and the tail behavior of the two distributions is about the same, the body of the distribution in Fig. 9(c) shifts to larger losses, and the probability of seeing zero-loss events nearly disappears. Calculating the differences between the loss distributions conditioned on different sets of input information can serve to quantify the value of more accurate AR forecasting tools for the study area.*

*"Figure 10 shows the spatial distribution of building losses from the February 2019 event, averaged across all Monte Carlo realizations and aggregated to the census block group level. Losses are concentrated along the banks of the Russian River with hotspots near Healdsburg, Guerneville, and the mouth of the river near the Pacific Ocean. These locations received warnings and evacuation orders before and during the storm event. While these particular communities are already known to have high vulnerability to flooding, the PARRA simulation results offer a new way to quantitatively prioritize investments in flood mitigation, from emergency communications to infrastructure projects to high-resolution modeling."*

[Figure]

*Revised Figure 9.*

[Figure]

*New Figure 10 (previously part of Figure 9).*

We would also like to note that due to a calculation error identified by Reviewer 1, the 2019 observed loss estimate now falls at the 89[th] percentile of the simulated distribution rather than the

98th percentile that was reported in the original version of the manuscript. This numerical correction, in addition to the more nuanced discussion of selection bias in the new version of Section 3.7.2, may alleviate some of the reviewer's concerns about the 2019 event being barely captured within the simulated distribution.

Regarding the reviewer's second concern about the description of the 2019 event case study, we recognize that we were not clear enough in describing its purpose. Our intent was not to reproduce the 2019 event exactly, but to use the mean and uncertainty of the simulated distributions to gain insight into the extremeness of this AR event, and that having an observed value fall in the upper tail of the simulated distribution is an interesting scientific result rather than an indication of poor model agreement. We have added discussion of the 2019 event in the new version of Sect. 3.2.2. We have also modified the manuscript in several places to remove all references to the 2019 event as a validation exercise and instead characterize it as a comparison, as summarized below.

- Line 11: The sentence "Evaluation of a case study AR event…" was removed and replaced with the following.
*"Individual component models are fit and validated against a historic catalog of AR events occurring from 1987-2019. Comparing simulated results from these component model implementations against observed historic ARs highlights what we can learn about the drivers of extremeness in different flood events by taking a probabilistic perspective."*

- Lines 198-206: We have removed these paragraphs and added the following new text.

*"Within Sonoma County we use the PARRA framework to examine the drivers and impacts of historical AR events. Each of the six subsections below corresponds with one of the pinch point variables defined in Sect. 2 and is divided into two parts. The first part of each subsection describes the user choices made to represent the study area within each component model. While we include many of the specific details related to fit and validation of these model implementations, the focus is on the overall workflow and how to functionally apply the PARRA framework. The second part of each subsection compares simulated Monte Carlo realizations of pinch point variables to observed data. These comparisons can be seen as a forensic reconstruction rather than an attempt to replicate the observed values. We focus on the new knowledge gained from the model implementations about how the observed values fall within the range of "what might have been."*

*"We present two types of case studies to showcase the breadth and depth of insights that are possible in a model-by-model analysis. For breadth, we compare and contrast observed vs. simulated precipitation values for four different AR events. We examine storms with varying AR intensity categories to determine which storms displayed "average" behavior for their category*

*and which exceeded predicted impacts. For depth, we focus the discussion for all other pinch points on a single Category 3 AR from February 2019, referred to as the February 2019 event. This event's recency combined with its severe impact mean that datasets unique to this event are available to compare many of the individual component model implementations against ground-truth observations, allowing a more focused analysis. Comparisons and results for additional storms can be found in the supplemental code release referenced at the end of the paper."*

- Line 373: The sentence "We examine three different strategies…" has been modified as follows.
*"Because the N-dimensional inundation pinch point variable $INUN$ contains significantly more data than we have generated thus far, we explore three strategies to compare the observed and simulated inundation maps."*

- Line 387: The sentence "Our LISFLOOD model…" has been modified as follows.
*"Our LISFLOOD model was able to reproduce the Sonoma GIS map with a critical success index of 67.77%, which indicates that the observed inundation is within the range of what we would reasonably predict given the observed hydrograph."*

- Line 408: The sentence "Because there was little…" has been modified as follows.
*"Because there was little available in the way of site-specific damage information, we used building safety as a proxy variable to facilitate investigation of observed vs. predicted damage."*

- Line 604: The sentence "We performed a step-by-step comparison…" has been modified as follows.
*"We performed step-by-step comparisons between each of these component models and ground-truth data from the case study AR events to show how the differences between the observed and simulated values produced new insights about what drove certain events towards extreme consequences and not others."*

- Line 607: The sentence "The total losses to residential homes…" has been removed.

**2.3)**
I have some comments below that can help improve the paper. I think the paper can be accepted after some minor revisions and clarifications.

Response: We would like to thank the reviewer for their recommendation, and for the time and effort spent reviewing this manuscript.

General Comments

**2.4)**

What would the PARRA framework provide a stakeholder as the estimated losses for the 2019 AR event?

Response: We thank the reviewer for this insightful question. Communicating what communities can gain from implementing the PARRA framework locally is central to its adoption. We believe that the most powerful result from our modeling approach is the ability to generate a loss exceedance curve (Fig. 11). The loss exceedance curve provides insight into the overall character of the study area's flood risk, as stated in Lines 610-615 of the manuscript. This is a novel result made possible by the PARRA framework's ability to analyze potential losses over large stochastic event records.

We believe the strengths of the PARRA framework are not necessarily in forecasting or reproducing losses for individual events, but rather in drawing relative comparisons between different scenarios and designing to performance-based targets. We have emphasized the utility of relative comparisons in the new version of Sect. 3.7.2 included in our response to comment 2.2, especially in the paragraph describing Fig. 9(c). We have modified the description of the mitigation exercise presented in Sect. 4.3 to better highlight the utility of the performance-based approach. We have removed the paragraph starting at Line 508 and replaced it with the following text:

*"A benefit of taking a performance-based approach is the ability to set a specified performance objective, such as loss reduction, and determine what changes can be made to the hazard, exposure, and vulnerability to reach that target. Working backwards to design a system that meets a set performance target is a powerful and unique capability of performance-based frameworks. Here we demonstrate the performance-based aspect of the PARRA framework through a hypothetical mitigation analysis. We define a target loss reduction threshold of reducing the AAL by half, and we assess the effectiveness of home elevation as a pathway to meet that threshold. We then quantify the effects of the system changes on the shape of the loss exceedance curve to highlight the framework's capability to prospectively assess events without historical precedent."*

We believe that this response, coupled with our response to comment 2.2 above, serves to additionally resolve the concerns and questions expressed by the reviewer in comments 2.5, 2.20, 2.33, and 2.34.

**2.5)**

The results show an average loss of about $25 million (Figure 9a), but the actual cost was $91.6 million. This actual cost is covered in the tail of the distribution provided by PARRA but is far from the mean of this distribution. Do the authors consider this an accurate assessment? Some

comments on how to interpret the results with their associated uncertainties, as well as how to interpret the acceptability of the results, would be helpful.

Response: Please see our responses to comments 2.2 and 2.4, where we discuss the accuracy of the case study assessment and stakeholder interpretation of the case study results, respectively.

Specific Comments / Technical Corrections

Section 1 Introduction

**2.6)**
Line 27: Pineapple express is not the only mechanism that brings ARs to California.

Response: We acknowledge the reviewer's suggestion and we have removed this sentence from the manuscript.

**2.7)**
Line 56: "... understanding climatology of ARs", see Espinoza et al. 2018 and Massoud et al. 2019 who aimed to understand AR climatology in a global context.

(a) Espinoza, Vicky, Duane E. Waliser, Bin Guan, David A. Lavers, and F. Martin Ralph. "Global analysis of climate change projection effects on atmospheric rivers." Geophysical Research Letters 45, no. 9 (2018): 4299-4308.

(b) Massoud, E. C., V. Espinoza, B. Guan, and D. E. Waliser. "Global climate model ensemble approaches for future projections of atmospheric rivers." Earth's Future 7, no. 10 (2019): 1136-1151.

Response: We thank the reviewer for providing these references, and we appreciate the additional information on how the global climatology of ARs will change in the future. However, because our work is focused regionally on California and because we do not consider the effects of a future climate, we could not find a location in the manuscript to include these additional citations.

Section 2 Framework Description

**2.8)**
Line 110: Is this theorem a version of Bayes theorem? How are they related?

Response: The total probability theorem and Bayes' theorem are both applications of the rules of conditional probability, albeit for different purposes. The total probability theorem, seen at Line 110 in the manuscript, states that $P(A) = \sum_{i=1}^{n} P(A|B_i) * P(B_i)$ for any set of mutually exclusive, collectively exhaustive events $B_i$ within the partitioned event space $B$. This is used to calculate the overall probability of event $A$ using information about the conditional probabilities of $A$ within different partitions of the event space. Bayes' theorem states that $P(A|B) * P(B) = P(B|A) * P(A)$. This is a way to use information about one event to update our estimate of the probability of another.

This work relies heavily on the total probability theorem, as it is the foundation of probabilistic risk analysis: we can only know the full spectrum of flood risk outcomes if we individually consider all possible events that could lead to flooding. Bayes' theorem is not as applicable for our use case.

**2.9)**
Line 143: Initially it seems that the AR category score (1-5) is used as input in the PARRA framework. It isn't until later in the manuscript that it becomes clear that AR max IVT and duration are used. The authors should clarify this earlier in the paper.

Response: We appreciate the reviewer identifying this source of confusion. We have revised the paragraph about ARs in Sect. 2.1 to read as follows.

*"The pinch point representing an atmospheric river event (AR) is characterized as a vector with two elements: the maximum recorded integrated water vapor transport (IVT) (kg·m$^{-1}$s$^{-1}$), and the duration (h) of sustained IVT exceeding 250 kg·m$^{-1}$s$^{-1}$. These were chosen as metrics of interest because of their connection to impacts. Based on maximum IVT and duration, the bivariate AR intensity scale proposed by Ralph et al. (2019) ranks ARs from 1–5 to qualitatively summarize their expected severity (from weak to exceptional) and potential consequences (from beneficial to hazardous). Category 1 ARs are classified as primarily beneficial storms, replenishing the water supply without causing adverse effects. Category 5 ARs are classified as primarily hazardous with a high likelihood of flooding and damage."*

**2.10)**
Line 147: Some precipitation can be from non-AR sources. Is this considered for the calculation of the precipitation submodule in the PARRA framework?

Response: Precipitation from non-AR sources is not included here, because we focus on precipitation that leads to floods and damaging impacts in Sonoma County, and the overwhelming majority of flood damage in Sonoma County is due to ARs (>99%, as referenced in Line 186 of the manuscript). We have added the following sentence to Line 148 to clarify our

scope: *"Only precipitation associated with ARs is included in this analysis."*

**2.11)**
Line 160: take out the word 'are'

Response: We have revised according to the reviewer's suggestion.

Section 3 Case Study: Sonoma County

**2.12)**
Line 235: Is there a citation that shows why WLS can be used to express the relationship between IVT/DUR and PRCP? This seems rather simplistic and not thorough enough to capture the estimated PRCP. According to Figure 4 there seems to be significant spread in these relationships. Perhaps the authors can explain why this choice was made.

Response: The use of a statistical regression to represent the precipitation component model was a subjective implementation choice specific to the Sonoma County case study, and we fit the regression coefficients with WLS instead of OLS because it more accurately captured the heteroskedasticity of the observed data. We believe that the WLS regression accurately captures the distribution of the observed precipitation record, as evidenced by the Q-Q plot in Fig. 4(c) and by the K-S test statistic reported at Line 249 of the manuscript that fails to reject the null hypothesis of different distributions at a 95% confidence level. We intentionally chose a simpler representation of precipitation for a few reasons. For example, we were able to explicitly parameterize the uncertainty, which would not have been possible with a dynamical precipitation model, and we were able to show a range of implementation complexity levels available to framework users. We have further discussed the benefits of stochastic vs. deterministic modeling and the subjective choices within the case study implementation in our responses to comments 2.29 and 2.30, respectively.

We also appreciate the reviewer calling attention to the scatterplots in Fig. 4. Because precipitation is a function of the bivariate statistical relationship between maximum IVT and duration, Figs. 4(a-b) are showing two "flattened" projections of the multidimensional space, and as a result the point clouds look fairly disperse. Based on the reviewer's comments we have added color scales to Figs. 4(a-b) representing the "flattened" dimension in each. Combined with the regression fit lines at multiple values that were included previously, we believe that this new visualization better shows that the bivariate WLS regression does quite well in capturing the spread of precipitation outcomes.

[Figure]

*Revised Figure 4.*

**2.13)**

Line 260: The mean of the distribution is way off here. Should this be a reason for concern? It seems that this methodology begins to break down for extreme AR events.

Response: We believe that the reviewer's concern about the observed vs. simulated comparison for the February 2019 event has been addressed by the additional contextualization we have presented in the new version of Section 3.2.2, as presented in our response to comment 2.2. We would also like to note that the Q-Q plot in Fig. 4(c) and the K-S test statistic reported at Line 249 are metrics of how well the distribution captures all extremes, not just the February 2019 event, and both metrics support the assertion that the observed precipitation distribution is well represented by the chosen component model implementation.

**2.14)**

Line 266: "... a suitable representation of reality", this is a subjective acceptance criterion, and the authors should note it as so.

Response: We acknowledge the reviewer's suggestion and we have removed this sentence from the manuscript.

**2.15)**

Line 282: Figure 6 is mentioned before Figure 5.

Response: We appreciate the reviewer noticing this inconsistency. We have renumbered the figures so that they are now introduced in numerical order.

**2.16)**

Line 332/Figure 5: The initial peak on Feb 26 is not captured. Can the authors provide some comments and reasoning behind this?

Response: We acknowledge that the early streamflow peak seen in Fig. 5(b) requires more contextualization. We have added a hyetograph of observed precipitation to the figure as shown below. We have also added additional text to the manuscript to contextualize both the early peak and the new hyetograph. The new text starts at Line 332 and reads as follows.

*"The complex shape of the observed streamflow timeseries in Fig. 5(b) is a function of the unique watershed response as well as the spatial and temporal heterogeneity of the input precipitation. By contrast, the simulated distribution is based on the unit hydrograph method, which assumes that the precipitation distribution is uniform and that all runoff enters the channel at a single location. This limits our ability to capture certain kinds of behavior, such as the early peak seen in the observed streamflow timeseries in Fig. 5(b). The early peak could be due to catchment processes that cause a lagged tributary response, input from direct surface runoff, spatial variation in precipitation intensity and duration, or any number of other mechanisms. We include the observed hyetograph to the top of the plot in Fig. 5(b) to show just one aspect of the natural variability that affects the observed timeseries.*

*"Despite the simplification imposed by the unit hydrograph method, though, many metrics of interest are reasonably characterized by the simulated timeseries. The observed peak streamflow ($1,130 \ m^3 s^{-1}$) is at the $43^{rd}$ percentile and the observed floodwave duration (81 h) is at the $63^{rd}$ percentile of the respective simulated distributions. Recall from Sect. 3.2 that the observed precipitation was anomalously high conditioned on the observed atmospheric conditions. We now note that while the observed streamflow may have been high relative to atmospheric conditions, it was in the middle of the predicted distribution conditioned on the observed*

*precipitation. Therefore we conclude that the hydrologic routing was likely not one of the physical processes contributing to the "extremeness" of the 2019 event."*

[Figure]

*Revised streamflow figure.*

*Caption: "Streamflow component model. All values are calculated in reference to USGS gage 11463500 (study area inlet). (a) Q–Q plot of observed vs. simulated peak streamflow (Qp) values for all events in the historic catalog. (b) Observed values from the historic catalog values vs. the fitted lognormal distribution for time to peak streamflow (tp). The dashed line indicates the observed time to peak value for the February 2019 event. (c) Distribution of simulated streamflow hydrograph realizations for the February 2019 event. The left axis represents observed hourly precipitation and the right axis represents streamflow. The observed hydrograph timeseries is shown as a black dashed line. The solid line represents the median of the simulated realizations, and the dark and light grey shaded areas represent the 50th and 90th percentile prediction intervals, respectively. The dark red horizontal line indicates the National Weather Service (NWS) flood flow for USGS gage 11463500.*

**2.17)**

Line 339: This comment is applicable for this section and for other sections. There are several choices that need to be made by the user, such as the LISFLOOD parameters. This raises the question of the PARRA method's applicability to other locations. Does the whole framework need to be re-calibrated with local data for other local case studies?

Response: We thank the reviewer for raising these concerns about both the implementation choices for the Sonoma County case study and the applicability of the PARRA framework to other locations. We have addressed the first concern by strengthening the distinction between the PARRA framework as a risk analysis tool and the proof-of-concept case study implementation of the framework within Sonoma County. In particular we have differentiated "pinch points" vs. "pinch point variables" and "component models" vs. "component model implementations," where the former is related to the generalized framework and the latter is related to the specific case study, respectively. We believe that understanding these differences is key to understanding the purpose of the manuscript, and we have expanded upon them as follows.

We have added the following paragraph at Line 139 to Sect. 2.1, "Pinch Point Variables."
*"The pinch points presented in Sect. 2 are conceptual descriptions of the intermediate system states between AR occurrence and flood loss where only a limited amount of information must be transferred to the next step. Pinch point variables are low-dimensional numerical vectors representing the information recorded at each pinch point (Garrick, 1984). The following paragraphs expand upon the conceptual pinch points and introduce the specific dimensions and measurement units that are used in this paper for each pinch point variable."*

We have added the following paragraph at Line 173 to Sect. 2.2, "Component Models."
*"Throughout this paper we make the distinction between component models, which have been presented thus far in a theoretical sense, and component model implementations. The component model implementations are the choices made by users of the PARRA framework about how a particular physical process will be represented, including what type of model to use (i.e., statistical vs. dynamical), the temporal and spatial resolution of analysis, etc. The state of atmospheric and hydrologic modeling is ever changing, and the "best" implementation choice depends on the modeler, the study area, and the intended end use (Baker et al., 2021). We have intentionally presented the PARRA framework in this section without tying the component models to specific implementations."*

We have added the following paragraph at Line 181.
*"The PARRA framework has scientific value as an internally consistent and logically sound structure to connect atmospheric phenomena to community-level impacts. It enables the communication of ideas and results across disparate research fields, isolates the uncertainty associated with different processes within the model chain, and introduces new avenues for interdisciplinary collaboration. By contrast, implementing the PARRA framework for any given*

*location imposes constraints, but also opens the door for practical insights. Site-specific implementations of the PARRA framework and component models are what quantify the probabilistic range of potential risk outcomes and generate actionable insights for stakeholders within case study communities.”*

To address the second concern, we elaborated upon the existing discussion of the framework's data needs in Sect. 5.3, "Validation Data." While the framework itself is a performance-based risk analysis tool that does not need calibration, the component model implementations that enable community-level insights do require local data to be meaningful at the local level. Some of the component model implementations presented here would be harder to move to new locations than others; for example, the statistical representation of precipitation with a WLS regression could be easily refit, but the calibrating the hydrodynamic inundation model to a new location would require much more time and effort. The changes and additions we have made to the manuscript to address the reviewer's concerns are outlined below.

- Line 562: The following sentences have been added.
*"The PARRA framework is a risk analysis tool that is globally applicable and can be used to assess AR flood risk at any scale. However, the implementation of the framework in any location will inherently be case-specific, and local insights require models calibrated to local conditions.”*

- Line 571: The sentence "As a consequence…" has been modified as follows.
*"As a consequence, moving from inundation to damage and from damage to loss are the most uncertain aspects of flood risk assessment due to large uncertainties in the physical mechanisms and the documented difficulties in validating against observed data (Apel et al., 2009; Gerl et al., 2016). Therefore these would be the hardest component models to implement in a new location if the PARRA framework is applied elsewhere.”*

- Line 580: The following sentence has been added.
*"In return, implementing the PARRA framework with fine-resolution local data provides communities with much more relevant information than they would be able to gain from a large-scale regional or global flood risk assessment.”*

**2.18)**
Line 352: There is no information on which surrogate model was used, and what the accuracy or efficiency of that surrogate model is. In general, there is very little information on this emulation method or how it is used. How can other readers re-produce or build on this analysis if this critical information is missing?

Response: We define and explain many aspects of the surrogate model in Lines 356-365 of the manuscript, which are restated here for convenience. We used the inverse distance weighting spatial interpolation method as our surrogate model to rapidly generate new inundation maps. The predictor variables are $Q_p$ (peak streamflow, $m^3s^{-1}$) and $t_p$ (time to peak streamflow, hrs) and the response surface is the full inundation map (depth estimates in meters at 925,000 grid cells). The hyperparameters of the model were fit by ten-fold cross-validation, the error metric was RMSE, and the fitted vertical accuracy was 3.5 cm.

Regarding the reviewer's question about reproducibility: in November 2021 we published extensive code, data, and documentation for the Sonoma County case study on Github, including step-by-step instructions to run every component model and reproduce all data figures, with a particular emphasis on the fit and calibration of LISFLOOD and the surrogate model. The code release is mentioned in the "Code and data availability" section at Line 621 and in the manuscript at Line 349. We have moved the sentence in the manuscript to Line 365 and more explicitly called out the significant effort we have invested into making this work reproducible. The new sentence is as follows:

*"Additional information, including data, documentation, and reproducible code, to replicate the fit and calibration of both LISFLOOD and the surrogate model can be found in the supplemental code release, which is referenced in the "Data and code availability" section at the end of this paper."*

**2.19)**
Line 440: On correction factors - Again, this seems like a subjective fix for applying the PARRA framework in this region. How can the framework be applied elsewhere using this methodology? Although the framework seems to be useful for Sonoma County, how can the authors show that the methodology can still be efficiently applied for other locations? Some ideas that address this question can be helpful in accepting the PARRA framework as a generally usable framework

Response: Please see our response to comment 2.17, where we discuss the distinction between the PARRA framework and the Sonoma County case study implementation as well as the applicability of the PARRA framework to other locations.

**2.20)**
Figure 9: The distribution just barely captures the observed event in its tail. As mentioned above, how is this result with its uncertainty reported to a manager or a stakeholder? What is the provided answer here?

Response: Please see our responses to comments 2.2 and 2.4, where we discuss the accuracy of the case study assessment and stakeholder interpretation of the case study results, respectively.

Section 4 Results

**2.21)**
Line 477: Equation 6 - Is this equation reported anywhere in the literature? Seems like another subjective criteria that the authors implement. There needs to be more information describing this choice.

Response: Eq. 6 is used frequently in risk analysis and is the standard equation for estimating average annual loss (AAL) from empirical data, though we recognize that we should have provided this context for the diverse readers of this manuscript. We now reference the books *Catastrophe Modeling: A New Approach to Managing Risk* (Grossi and Kunreuther, 2005) and *Probabilistic Seismic Hazard and Risk Analysis* (Baker et al., 2021) on Line 478 to provide background on the concept and the calculation of AALs.

**2.22)**
Line 482: The AAL is an interesting concept to describe the average annual losses. However, it is known that this region experiences significant swings between wet and dry years. Is it feasible for the authors to calculate what the AAL is for wet vs dry years?

Response: The AAL is most often used in catastrophe modeling and probabilistic risk analysis as a tool to make long-range decisions about the future: whether to insure, mitigate, develop, etc. at one location versus another, and whether that decision will be profitable over a time horizon of decades to centuries. Because it is only a single value it does not provide much insight into the character of the risk, i.e., whether a location is historically dominated by lots of small losses vs. infrequent large losses, or whether there is high seasonal or interannual variability in the loss record. The simplicity of the metric, though, is what makes it useful for cost-benefit decisions. We have therefore decided not to calculate a conditional AAL for wet vs. dry years because in order to use it for decisionmaking we would need information we do not yet have about the future. For assessing variations from year to year the presented loss exceedance curve is the most relevant result.

**2.23)**
Line 487: What are the uncertainties around these estimates? Do the authors provide this?

Response: We have revised Line 482 to report both a mean estimate and uncertainty bounds. Please note that due to the numerical changes made in response to Reviewer 1 the mean estimate has been revised downward from the original version of the manuscript.

*"The mean AAL estimated from the stochastic record for AR-induced flood losses to residential structures is $111 million, with 90% confidence that it lies between $93 and $133 million."*

**2.24)**
Line 522: the word 'the' is duplicated

Response: We have revised according to the reviewer's suggestion.

**2.25)**
Line 524: 'Expected benefits' - See Massoud et al. 2018, who did a similar analysis for groundwater and investigated how changes to decisions in managing water resources can impact expected changes to groundwater storage. Studies like this are starting to populate the literature.

(a) Massoud, Elias C., Adam J. Purdy, Michelle E. Miro, and James S. Famiglietti. "Projecting groundwater storage changes in California's Central Valley." Scientific reports 8, no. 1 (2018): 1-9.

Response: We thank the reviewer for suggesting this reference. We agree with the reviewer that investigations of decision-making related to future water resource management and the benefits supplied by different water management strategies are certainly becoming more prevalent in the literature. However, we do not see a strong link with this publication to the discussion of our framework, and we did not find a location in the manuscript where a citation would be relevant.

**2.26)**
Line 529: Take out the words 'is of the 2019'.

Response: We have revised according to the reviewer's suggestion.

Section 5 Discussion

**2.27)**
Line 535: I would argue that these insights are helpful for planners, managers, and engineers, yet not so helpful for purely scientific investigation since many choices in the framework are purely subjective. I think it is important for the authors to make this clear throughout the paper.

Response: We are glad the reviewer considers our insights to be helpful for planners, managers, and engineers. We believe that the PARRA framework is a novel contribution to the field and that it has intrinsic scientific value as a physically based, modular, probabilistic, and prospective structure to connect atmospheric phenomena to community-level impacts, as stated in Lines 43-

50 and Lines 82-83. We have added new text at Line 616 in the manuscript to highlight these scientific contributions as follows.

*"While the case study showed examples of the specific insights that can be gained from implementing the component models for a community risk assessment, the theory and scientific merit of the PARRA framework stand on their own, independent from the specific benefits and tradeoffs inherent in any local implementation. We have proposed a new method for the structured assessment of AR-driven flood risk that is physically based, modular, probabilistic, and prospective."*

We have additionally discussed the distinction between the PARRA framework and the Sonoma County case study implementation in our response to comment 2.17, and we have discussed the subjective choices within the case study implementation in our response to comment 2.30.

**2.28)**
Line 543: Another process that can matter here is the role of sequential ARs (i.e., multiple ARs occurring sequentially), something to consider for 'future directions'.

Response: We appreciate the reviewer's suggestion of a direction for future research. We now mention the role of sequential and compounding events as a potential area for further exploration in the new version of Sect. 3.7.2, which is included in our response to comment 2.2.

**2.29)**
Line 557: Yes, but what did this do to the expected accuracy of capturing the relationships? The framework is trading potential accuracy and confidence for computational efficiency. This introduces even more uncertainty. The authors should state this.

Response: We agree that there are many potential component model implementations, each with different strengths and tradeoffs, that could have been used to capture relationships between variables at each step. However, the purpose of the PARRA framework is not to model one event perfectly, but instead to consider a stochastic range of potential events. We are presenting a modeling approach geared towards a different analysis procedure (Monte Carlo simulation) and end goal (probabilistic risk assessment) than that of deterministic modeling. We have expanded the discussion of implementation considerations in Sect. 5.2, "Framework Implementation," to better explain the value of the performance-based probabilistic approach.

- Line 548: The following paragraph has been added.
*"The PARRA framework represents one end of the continuum between stochastic and deterministic modeling. It will not perform as well as a high-fidelity multi-scale physics model calibrated to a given set of input forcings for a specific scenario event, and it is not intended to*

*replace existing models designed for that use case. However, it is impossible to scale the granular analysis performed by deterministic models to produce a probabilistic estimate of flood risk across a range of potential AR events (Apel et al., 2004; Savage et al., 2016). The PARRA framework thus serves a fundamentally different purpose within the literature of risk—if deterministic modeling gives us the best possible representation of a single tree, then the PARRA framework aims to characterize the shape and scale of the entire forest."*

- Line 548: The sentence "Another strength of the PARRA framework…" has been modified as follows.
*"A key functionality of the PARRA framework is its ability to track and quantify uncertainty across multiple component models."*

- Line 554: The sentence "We made several implementation decisions…" has been modified as follows.
*"We came up with several practical solutions to minimize the computational expense of the PARRA framework and bring it into the range of procedural feasibility without compromising accuracy."*

**2.30)**
Line 582: Component Model Alternatives - This is where some of the subjective choices of the framework can be replaced with more objective choices, and therefore can make the framework more sound for scientific analysis.

Response: We agree with the reviewer that the choices made to implement the PARRA framework were subjective choices made to optimize a specific set of priorities. However, choosing priorities (speed vs. complexity, local insights vs. broader trends, etc.) and making implementation choices that optimize for those priorities will always be a subjective process, and we believe there is no objective or "correct" implementation choice for any component model in the PARRA framework. We believe that the PARRA framework is sound for scientific analysis as presented, and we have added the following paragraph to Sect. 5.4, "Component Model Alternatives" at Line 582 to highlight this.

*"All models are imperfect representations of the physical world, and there will always be some nuance lost when moving from theory (the framework) to practice (the implementation). There are multiple possible methods to characterize some of the pinch points that would improve fidelity to the underlying physical processes but would increase computational demand and therefore constrain the representation of the true uncertainty. These inherent tradeoffs between different types of error are unavoidable, but point to a major strength of the PARRA framework: that the user is able to explicitly define their own optimization criteria and choose the component*

*model implementations that best suit their personal expertise, resource constraints, and end goals."*

**2.31)**
Line 589: the word 'the' is duplicated

Response: We have revised according to the reviewer's suggestion.

**2.32)**
Line 591: the word 'underlying' is duplicated

Response: We have revised according to the reviewer's suggestion.

Section 6 Conclusions

**2.33)**
Line 594: Is it possible/feasible to test another case study event that the PARRA framework accurately estimates the damages for? This can help show case the value of the PARRA framework even more than just showing the one case study from 2019 that was barely captured in the tail of the distribution.

Response: Please see our responses to comments 2.2 and 2.4, where we discuss the accuracy of the case study assessment and stakeholder interpretation of the case study results, respectively.

**2.34)**
Line 608: '... event fell within the expected probabilistic range ...', In the tail of the distribution. It was barely captured. The authors should be careful with how they communicate the accuracy of the provided result.

Response: Please see our responses to comments 2.2 and 2.4, where we discuss the accuracy of the case study assessment and stakeholder interpretation of the case study results, respectively.